# Learning Synthetic Environments and Reward Networks for Reinforcement Learning

**Fabio Ferreira** [1]  **Thomas Nierhoff** [1]  **Andreas Sälinger** [1]  **Frank Hutter** [1,2]
[1] University of Freiburg [2] Bosch Center for Artificial Intelligence
{ferreira, fh}@cs.uni-freiburg.de

## Abstract

We introduce *Synthetic Environments* (SEs) and *Reward Networks* (RNs), represented by neural networks, as proxy environment models for training Reinforcement Learning (RL) agents. We show that an agent, after being trained exclusively on the SE, is able to solve the corresponding real environment. While an SE acts as a full proxy to a real environment by learning about its state dynamics and rewards, an RN is a partial proxy that learns to augment or replace rewards. We use bi-level optimization to evolve SEs and RNs: the inner loop trains the RL agent, and the outer loop trains the parameters of the SE / RN via an evolution strategy. We evaluate our proposed new concept on a broad range of RL algorithms and classic control environments. In a one-to-one comparison, learning an SE proxy requires more interactions with the real environment than training agents only on the real environment. However, once such an SE has been learned, we do not need *any* interactions with the real environment to train new agents. Moreover, the learned SE proxies allow us to train agents with fewer interactions while maintaining the original task performance. Our empirical results suggest that SEs achieve this result by learning informed representations that bias the agents towards relevant states. Moreover, we find that these proxies are robust against hyperparameter variation and can also transfer to unseen agents.

## 1 Introduction

Generating synthetic data addresses the question of what data is required to achieve a rich learning experience in machine learning. Next to increasing the amount of available data, synthetic data can enable higher training efficiency that opens up new applications for Neural Architecture Search (Such et al., 2020), may improve algorithm analysis or facilitate custom datasets (Jhang et al., 2020).

In this paper, we consider learning neural synthetic data generators for Reinforcement Learning (RL). We investigate the question of whether we can learn a synthetic Markov Decision Process of a real (target) environment which is capable of producing synthetic data to allow effective and more efficient agent training, that is, to achieve similar or higher performance more quickly compared to when training purely on the real environment. When learning to produce both states and rewards, we refer to these neural network proxies as *synthetic environments* (SEs). Additionally, we investigate the same question for learning reward proxies that do not learn about the state dynamics and which we refer to as a *Reward Networks* (RNs).

We depict our procedure in Figure 1 which resembles a bi-level optimization scheme consisting of an outer and inner loop. The inner loop trains the agent on an SE or RN. Since our method is agnostic to both domain and agent, we can interchangeably adopt standard RL algorithms in the inner loop. In the outer loop, we assess the agent's performance by evaluating it on the real environment; we then take the collected reward as a score to update the SE's or RN's neural parameters used in the inner loop. In this way, the SE/RN is gradually updated such that an agent being trained on it, scores higher on a real environment. For the outer loop we use Evolution Strategies (Rechenberg, 1973; Salimans et al., 2017) with a population of SE/RN parameters.

After discussing related work (Section 2), we make the following contributions:

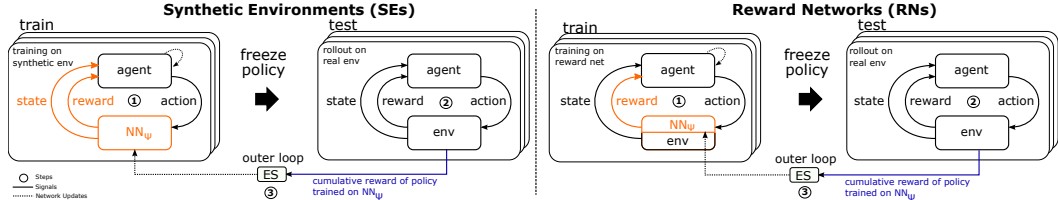

Figure 1: We use an agent-agnostic meta-learning approach to learn neural proxy RL environments (left) and reward networks (right) for a target task. In the inner loop, we train RL agents on the proxy and use the evaluation performance on the target task in the outer loop to evolve the proxy.

- We introduce synthetic environments (Section 3), a novel concept that focuses on the environment instead of agent learning using a bi-level optimization scheme, which is guided purely by the agent performance. This concept goes beyond the usual learning of a one-time internal environment model inside an agent (such as in Dyna (Sutton, 1990)).

- As a sub-problem of SEs, we investigate reward networks (Section 4), contrasting several types of potential-based reward shaping (Ng et al., 1999) variants.

- We show that it is possible to learn SEs (Section 5) and RNs (Section 6) that, when used for agent training, yield agents that successfully solve the Gym tasks (Brockman et al., 2016) CartPole and Acrobot (SEs), as well as Cliff Walking, CartPole, MountainCarContinuous, and HalfCheetah (RNs).

- We show that SEs and RNs are efficient and robust in training agents, require fewer training steps compared to training on the real environment and are able to train unseen agents

- We report empirical evidence showing that SEs and RNs achieve their efficiency gains for training new agents through condensed and informed state and reward representations.

Overall, we find it noteworthy that it is actually possible to learn such proxies, and we believe our research will improve their understanding. Since these learned proxies can train agents quickly and transfer to unseen agents, we believe this work might open up possible avenues of research in RL. Possible future applications include cheap-to-run environments for AutoML (Hutter et al., 2019) or for robotics when training on targets is expensive, as well as agent and task analysis, or efficient RL agent pre-training. Our PyTorch (Paszke et al., 2019) code and models are made available publicly.[1]

## 2 RELATED WORK

**Synthetic Environments**   In the context of RL, learning synthetic environments is related to model-based RL (MBRL) where the dynamics model can be viewed as an SE. In MBRL, one jointly learns both a dynamics model and a policy as in Dyna (Sutton, 1990) or an existing dynamics model is used with planning methods to learn a policy (Silver et al., 2017; Moerland et al., 2020). Our work does not involve planning, nor does it use supervised learning for the dynamics model and it does not mix synthetic and real data during policy learning. Instead, we use purely synthetic data to train agents similar to World Models (Ha & Schmidhuber, 2018) but use the cumulative rewards from the real environment in a bi-level optimization for learning our model jointly with our agent. For a more extensive discussion on the differences between our work and model-based RL, we refer the reader to Appendix D.

Analogous to learning SEs is Procedural Content Generation (Togelius et al., 2011) and Curriculum Learning that concerns automatically selecting (Matiisen et al., 2020) or generating the content of training environments (Volz et al., 2018; Shaker et al., 2016; Wang et al., 2019; Cobbe et al., 2020), with the closest work being Generative Playing Networks (GPNs) (Bontrager & Togelius, 2020). GPNs learn an environment generator that creates increasingly difficult SEs according to what a critic's value function estimates as challenging. While GPNs are a method to generate environment curricula, our approach studies learning an SE for effective and efficient agent training by compressing the relevant information into a single model of the environment. We also use a more generally

---

[1]https://github.com/automl/learning_environments

applicable objective that does not rely on actor-critic formulations but purely on the achieved cumulative reward. Less related areas are methods like domain randomization (Tobin et al., 2017) that generate environments regardless of agent performance or minimax methods that adversarially (instead of cooperatively) generate environments based on the difference between the reward of two competing agents that are trying to solve the generated environment (Dennis et al., 2020).

Related to our work, and inspiring it, are Generative Teaching Networks (Such et al., 2020). While we similarly use a bi-level optimization to learn a synthetic data generator, our approach is different in central aspects: we use Evolution Strategies to avoid the need for explicitly computing Hessians, we do not use noise vectors as input to our SEs, and we target sequential decision-making problems instead of supervised learning.

**Reward Networks**  Reward shaping concerns the question of how to enhance the reward signal to allow agents to be trained more effectively or efficiently. Common learned reward shaping approaches are curiosity or count-based exploration (Pathak et al., 2017; Burda et al., 2019; Singh et al., 2010; Bellemare et al., 2016; Tang et al., 2017). Others achieve reward shaping with prior knowledge through expert demonstrations (Judah et al., 2014; Brys et al., 2015; Ibarz et al., 2018). In contrast to our work, these contributions all apply a single-level optimization. When using a bi-level optimization, the reward shaping function is usually learned in the outer loop while the policy using the learned rewards is optimized in the inner loop. Here, one way is to meta-learn the parameterization of reward functions (Faust et al., 2019; Hu et al., 2020; Jaderberg et al., 2019). Another way is to learn a neural network that resembles the reward function.

While learning full synthetic environments is entirely novel, there exists prior work on learning reward shaping networks. The most related works are (Zheng et al., 2018) for single tasks, (Zou et al., 2019) for entire task distributions, or (Zheng et al., 2020) that additionally take into account the entire lifetime of an agent to learn a "statefulness across episodes"-reward function. Despite the similarities, some noteworthy differences exist. Importantly, the approaches in (Zheng et al., 2018; Zou et al., 2019) are not agent-agnostic, making it less straightforward to exchange agents as in our work. Moreover, the transferability of learned shaped rewards is studied only limitedly (Zheng et al., 2018), not at all (Zou et al., 2019) or only for grid world-like environments (Zheng et al., 2020).

## 3   LEARNING SYNTHETIC ENVIRONMENTS

**Problem Statement**  Let $(\mathcal{S}, \mathcal{A}, \mathcal{P}, \mathcal{R})$ be a Markov Decision Process (MDP) with the set of states $\mathcal{S}$, the set of actions $\mathcal{A}$, the state transition probabilities $\mathcal{P}$ and the immediate rewards $\mathcal{R}$ when transitioning from state $s \in \mathcal{S}$ to the next state $s' \in \mathcal{S}$ through action $a \in \mathcal{A}$. The MDPs we consider are either human-designed environments $\mathcal{E}_{real}$ or learned synthetic environments $\mathcal{E}_{syn,\psi}$ (SE) represented by a neural network with parameters $\psi$. Interfacing with the environments is identical in both cases, i.e. $s', r = \mathcal{E}(s, a)$. The crucial difference is that for SEs, the state dynamics and rewards are learned. The main objective of an RL agent when acting on an MDP $\mathcal{E}_{real}$ is to find a policy $\pi_\theta$ parameterized by $\theta$ that maximizes the cumulative expected reward $F(\theta; \mathcal{E}_{real})$. We consider the following bi-level optimization problem: find the parameters $\psi^*$, such that the agent policy $\pi_\theta$ parameterized with $\theta$ that results from training on $\mathcal{E}_{syn,\psi^*}$ achieves the highest reward on a target environment $\mathcal{E}_{real}$. Formally that is:

$$\psi^* = \arg\max_{\psi} \quad F(\theta^*(\psi); \mathcal{E}_{real})$$
$$\text{s.t.} \quad \theta^*(\psi) = \arg\max_{\theta} \quad F(\theta; \mathcal{E}_{syn,\psi}). \tag{1}$$

We use standard RL algorithms for optimizing the agents on the SE in the inner loop. Although gradient-based optimization methods can be applied in the outer loop, we chose Natural Evolution Strategies (Wierstra et al., 2008) (NES) to allow the optimization to be independent of the choice of the agent in the inner loop and to avoid potentially expensive, unstable meta-gradients (Metz et al., 2019). Additional advantages of NES are that it is better suited for long episodes, sparse or delayed rewards, and parallelization (Salimans et al., 2017).

**Algorithm**  We now explain our method. The overall scheme is adopted from (Salimans et al., 2017) and depicted in Algorithm 1. It consists of an Evolutionary Strategy in the outer loop

to learn the SE and an inner loop which trains RL agents. The performances of the trained agents are then used in the outer loop to update the SE. We instantiate the population search distribution as a multivariate Gaussian with mean 0 and a fixed covariance $\sigma^2 I$. The main difference to (Salimans et al., 2017) is that, while they maintain a population over agent parameter vectors, our population consists of SE parameter vectors. Moreover, our approach involves two optimizations (the agent and the SE parameters) instead of one (agent parameters).

Our algorithm first stochastically perturbs each population member $i$ which results in $\psi_i$ (Line 5). Then, a new randomly initialized agent is trained on the SE parameterized by $\psi_i$ for $n_e$ episodes (L6). The trained agent with fixed parameters is then tested across 10 episodes on the real environment (L7), yielding the average cumulative reward which we use as a score $F_{\psi,i}$. Finally, we update $\psi$ with a stochastic gradient estimate based on all member scores (L8). We use a parallelized version of the algorithm and an early stopping heuristic to stop in fewer than $n_e$ episodes when progress plateaus (more in Appendix A.1).

---

**Algorithm 1:** Learning Synthetic Env. with NES

1 Input: initial SE parameters $\psi$, real environment $\mathcal{E}_{real}$, NES noise std. dev. $\sigma$, number of episodes $n_e$, population size $n_p$, NES step size $\alpha$
2 **repeat**
3    **foreach** *member of pop.* $i = 1, 2, \ldots, n_p$ **do**
4       $\epsilon_i \sim \mathcal{N}(0, \sigma^2 I)$
5       $\psi_i = \psi + \epsilon_i$
6       $\theta_i = \text{TrainAgent}(\theta_i, \mathcal{E}_{syn,\psi_i}, n_e)$
7       $F_{\psi,i} = \text{EvaluateAgent}(\theta_i, \mathcal{E}_{real})$
8    Update SE: $\psi \leftarrow \psi + \alpha \frac{1}{n_p \sigma} \sum_i^{n_p} F_i \epsilon_i$
9 **until** $n_o$ *steps*

---

## 4 LEARNING REWARD NETWORKS

Learning both the state dynamics and rewards can be challenging (see Section 7). To reduce computational complexity but still be able to achieve training efficiency, one can therefore reduce the problem formulation to only learn (to augment) the rewards and make use of the real environment for the state dynamics (and original rewards) as illustrated in Figure 1. We reuse the formulation of $\mathcal{E}_{syn}$ and $\mathcal{E}_{real}$ from the problem statement (Eq. 1) and describe learning reward networks as getting the next state and reward $s', r = \mathcal{E}_{real}(s, a)$ and a corresponding synthetic reward $r_{syn} = \mathcal{E}_{rn,\psi}(s, a, s')$ with a (reward) environment represented by a neural network $\psi$. In the domain of *reward shaping*, $r_{syn}$ is usually added to $r$ to generate an augmented reward $r_{aug} = r + r_{syn}$. A popular approach in reward shaping is potential-based reward shaping (PBRS) (Ng et al., 1999), which defines $\mathcal{E}_{rn}$ as $\mathcal{E}_{rn,\psi}(s, a, s') = \gamma\Phi(s') - \Phi(s)$ with a potential function $\Phi : \mathcal{S} \rightarrow \mathbb{R}$ and a discount factor $\gamma \in (0, 1]$. We adopt PBRS for finding $\mathcal{E}_{rn,\psi}$, represent $\Phi$ as a neural network with parameters $\psi$ and use Algorithm 1 to learn $\psi$. PBRS has the useful property of preserving optimality of policies (Ng et al., 1999) but we also introduce additional variants of $\mathcal{E}_{rn,\psi}$, listed in Table 1, which are motivated by PBRS but are not typically referred to as PBRS as they may not have such guarantees. These variants are motivated by the idea that fewer constraints may enable better reward learning (e.g., in case the difference of potential functions imposes a too strong constraint on the hypothesis space). For simplicity, we refer to the output of Reward Networks (RNs) as $r_{aug}$.

We now point out the differences between the RN and the SE case. When training on SEs, we use a heuristic for determining the number of required training episodes since both rewards and states are synthetic (see Appendix A.1). In the RN case, the real states allow us to directly observe episode termination, thus such a heuristic is not needed. Another difference to SEs lies in the

Table 1: Overview of Reward Network variants.

| Reward Network | $r_{aug}$ |
|---|---|
| Additive Potential RN (Ng et al., 1999) | $r + \gamma\Phi(s') - \Phi(s)$ |
| Exclusive Potential RN | $\gamma\Phi(s') - \Phi(s)$ |
| Additive Non-Potential RN | $r + \Phi(s')$ |
| Exclusive Non-Potential RN | $\Phi(s')$ |

*EvaluateAgent* function of Algorithm 1: Optimizing SEs for the maximum cumulative reward automatically yielded fast training of agents as a side-product, likely due to efficient synthetic state dynamics. Since we do not learn state dynamics in RNs and motivated by the efficiency gains, we now directly optimize the number of training steps $n_{tr}$ the *TrainAgent* function requires to solve the real environment when being trained on an RN. Additionally, we include the cumulative reward threshold of the real environment $C_{sol}$ and the cumulative reward at the end of training $C_{fin}$, yielding the final objective $n_{tr} + w_{sol} * max(0, C_{sol} - C_{fin})$ (to be minimized) where $w_{sol} \in \mathbb{R}_{>0}$ gives

us control over efficiency and effectiveness an RN should achieve. In Appendix B.4, we study the influence of alternative RN optimization objectives.

## 5 EXPERIMENTS WITH SYNTHETIC ENVIRONMENTS

**Experimental Setup** So far we have described our proposed method on an abstract level and before we start with individual SE experiments we describe the experimental setup. In our work, we refer to the process of optimizing for suitable SEs with Algorithm 1 as *SE training* and the process of training agents on SEs as *agent training*). For both SE and agent training on the discrete-action-space CartPole-v0 and Acrobot-v1 environments, we use DDQN (van Hasselt et al., 2016). We also address generalization from an algorithmic viewpoint by varying the agent hyperparameters during SE training (Line 6) after each outer loop iteration. Also, we study how robustly found SEs can train agents under varying agent hyperparameters and how well they transfer in training unseen agents. For studying transferability of SEs, we use Dueling DDQN (Wang et al., 2016) and TD3 (Fujimoto & H. Hoof, 2018). TD3 is chosen because it does not solely rely on Q-Learning and is an algorithm of the actor-critic family. Both our tasks employ a discrete action space and to be able to use TD3 we use a Gumbel-Softmax distribution (Jang et al., 2017).

We wrap our algorithm in another outer loop to optimize some of the agent and NES HPs with the multi-fidelity Bayesian optimization method BOHB (Falkner et al., 2018) to identify stable HPs. The optimized HPs are reported in Table 3 in the appendix. We did not optimize some of the HPs that would negatively affect run time (e.g., population size, see Table 4 in the appendix). After identifying the agent and NES HPs, we removed BOHB and used the found HPs for SE training. Once SEs were found, we reverted from specific agent HPs that worked best for training SEs to default agent HPs as reported in Table 5 (appendix) to test out-of-the-box applicability.

**Feasibility** After introducing the core concept of learning SEs, we now investigate its efficacy of learning SEs for the CartPole and Acrobot tasks. First, we identified stable DDQN and NES hyperparameters. Then, we ran Algorithm 1 with 16 population members for 200 NES outer loop iterations to generate the SEs. We depict the result in Figure 2, which shows the average of the 16 members' evaluation scores (given by *EvaluateAgent* in Algorithm 1) as a function of the NES outer loop iterations for multiple NES optimization runs (thin lines). The thick solid lines show the average of these. Notice that 50 NES iterations are often sufficient to find SEs that teach agents to solve a task (for more details, we refer the reader to the Appendix Section A).

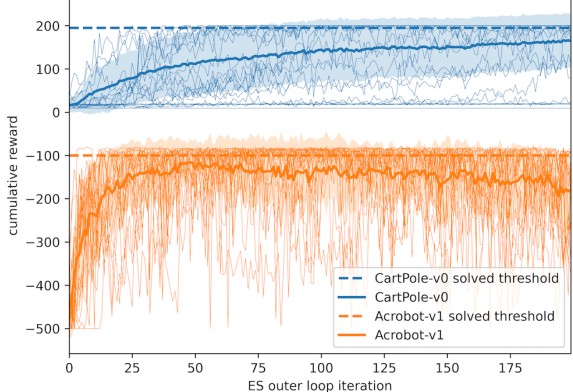

Figure 2: Multiple NES runs of Alg. 1 for CartPole (top) and Acrobot (bottom) which can teach agents the task effectively. Each thin line is the average of 16 worker scores from *EvaluateAgent* of such a run.

**Performance of SE-trained Agents** As we have shown in the last section, learning SEs that can teach agents the CartPole and Acrobot task is feasible with the proposed concept. Now, we investigate how efficient and hyperparameter-sensitive the generated SEs are. For this, we distinguish three different cases: 1) *train: real*, 2) *train: synth., HPs: varied*, and 3) *train: synth.,*

Table 2: Agent hyperparameter ranges sampled from for training SEs and agents.

| Agent hyperparam. | Value range | log scale |
|---|---|---|
| learning rate | $[10^{-3}/3, 10^{-3} * 3]$ | ✓ |
| batch size | $[42, 384]$ | ✓ |
| hidden size | $[42, 384]$ | ✓ |
| hidden layer | $[1, 3]$ | x |

*HPs: fixed*. The first is our baseline for which we train agents only on the real environment without any involvement of SEs. The second case denotes SEs for which we randomly sampled agent hyperparameters (HPs) from the ranges in Table 2 before each *TrainAgent* call *during SE training*. Lastly, the third case is similar to the second except that we did not randomly sample agent HPs but rather used the optimized ones reported in Table 3 which we kept *fixed during SE training*.

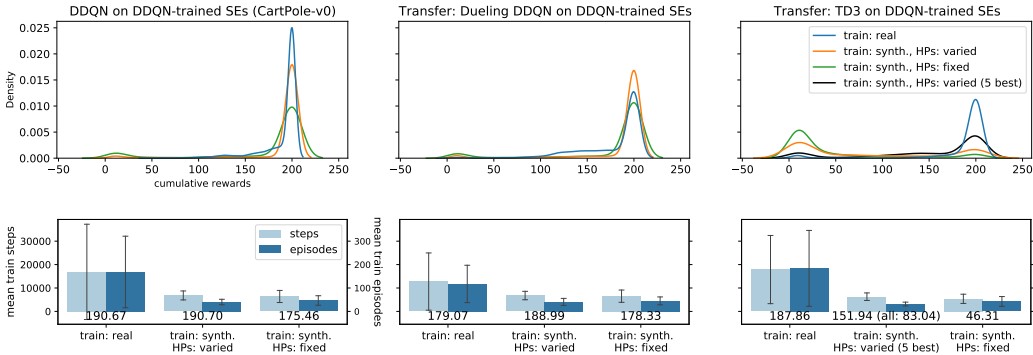

Figure 3: **Top row**: Densities based on each 4000 cumulative test rewards collected by DDQN (left), Dueling DDQN (center), and discrete TD3 (right) agents on CartPole. We show three settings: agents trained on a real environment without any involvement of SEs (blue, baseline), on SEs where the agent HPs were fixed during SE training (green), and on SEs when agent HPs were varied during SE training (orange). After training, the evaluation is always done on the real environment. In all three settings, we randomly sample the agent HPs before agent training. **Bottom row**: Average train steps and episodes corresponding to the densities along with the mean reward (below bars). When training on SEs, we train 30-65% faster compared to training on the real environment.

We first trained 40 SEs for each of the cases 2) and 3); then, we evaluated all approaches with the following procedure: on each SE we trained 10 DDQN agents with randomly sampled HPs according to Table 2. After agent training, we evaluated each agent on the real environment across 10 test episodes, resulting in 4000 evaluations. For our baseline, where no SE is involved, the procedure was identical except that we used 40 *real* environment instantiations instead of SEs to obtain the same number of evaluations. We then used the cumulative rewards from the test episodes for generating the densities in Figure 3 and report the average episodes and training steps required until the heuristic for stopping agent training on SEs/real environment is triggered (Appendix A.1).

Training DDQN agents on DDQN-trained SEs without varying the agent's HPs (green) during SE training clearly yielded the worst results. We attribute this to overfitting of the SEs to the specific agent HPs.[2] Instead, when varying the agent HPs (orange) during SE training, we observe that the SEs are consistently able to train agents using ∼60% fewer steps (6818 vs. 16887) on average while being more stable than the baseline (fewer train steps / lower episode standard deviations and little mass on density tails). Lastly, the SEs also show little sensitivity to HP variations.

**Transferability of SEs** In the previous section we evaluated the performance of DDQN-trained SEs in teaching DDQN agents. However, since many environment observations are needed for SE training, it would be beneficial if DDQN-trained SEs are capable of effectively training unseen agents as well. To study this question, we reuse the DDQN-trained SEs from above but now train Dueling DDQN agents on the SEs. From Figure 3 (top and bottom center) we conclude that the transfer to the Dueling DDQN agent succeeds and it facilitates a ∼50% faster (6781 vs. 12745 train steps), a more effective (higher reward), and noticeably more stable training (lower std. dev.; smaller tails) on average when compared to the baseline (orange vs. blue). As before, not varying the DDQN agent HPs during SE training reduces transfer performance (green).

We also analyzed the transfer from Q-Learning to a discrete version of TD3 (see Section 3). The result shown in Figure 3 (top right) indicates a limited transferability. We believe this may be due to the different learning dynamics of actor-critics compared to learning with DDQN. However, we also noticed that individual SE models can consistently train TD3 agents successfully while remaining efficient. To show this, we repeated the evaluation procedure from Figure 3 individually for each SE model (see Appendix F.1.1) and selected five SE models that yielded high rewards when training TD3 agents. As a consequence, we can again significantly improve the performance (black curve) with a ∼65% speed-up (6287 vs. 17874 train steps; 187 vs. 152 reward) over the baseline.

---

[2]We note that even in the "HPs: varied" case we are not evaluating whether SEs extrapolate to new HP ranges but rather how sensitive SEs react to HP variation within known ranges.

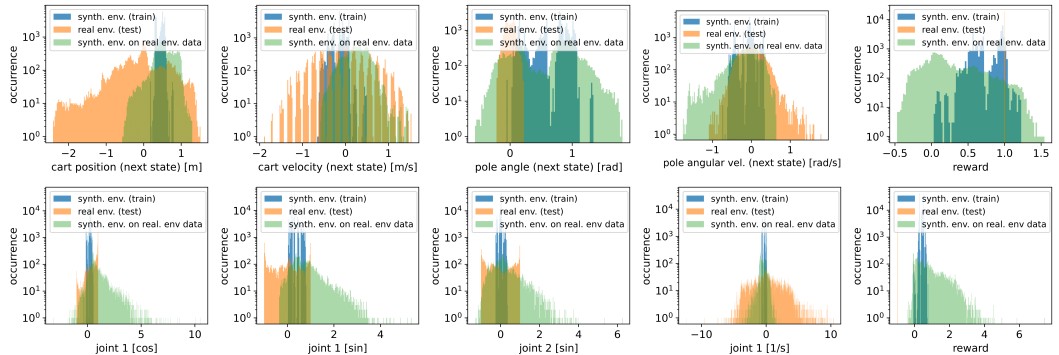

Figure 4: **Top row:** Histograms of next state $s'$ and reward $r$ produced by 10 DDQN agents when trained on a CartPole SE (blue) and afterwards tested for 10 episodes on a real environment (orange). We also show the SE responses when fed with real environment data seen during testing (green). **Bottom row:** The same for Acrobot (subset of state dimensions shown due to brevity).

We executed these experiments with similar results for the Acrobot task (solved with a reward $\geq$-100, see Figure 7 in the appendix). In short, we find that the transfer to Dueling DDQN facilitates a $\sim$38% faster training on average (18376 vs. 29531 steps, mean reward: -111). The discrete TD3 transfer shows a $\sim$82% (14408 vs. 80837 steps, five best models) speed-up but many of the runs also fail (reward: -343) which partly may be due to a sub-optimal hyperparameter selection. Simultaneously, we see that SE-learned policies in the Dueling DDQN transfer case are substantially better on average than those learned on the real environment. All in all, we believe these results are intriguing and showcase the potential of our concept to learn proxy RL environments.

**Analyzing SE Behavior** In the following, we try to illuminate the inner workings to explain the efficacy of the learned SEs. We exploit CartPole's and Acrobot's small state space and visualize an approximation of the state and reward distributions generated by agents trained on SEs and evaluated on real environments. First, we trained 10 DDQN agents on a randomly selected SE with default HPs and logged all $(s, a, r, s')$ tuples. Second, we evaluated the SE-trained DDQN agents on the real environment for 10 test episodes each and again logged the tuples. Lastly, we visualized the histograms of the collected next states and rewards in Figure 4. We observe distribution shifts between the SE and the real environment on both tasks, indicating the agent encounters states and rewards it has rarely seen during training, yet it can solve the task (reward: 199.3 on CartPole and -91.62 on Acrobot). Moreover, some of the synthetic *state* distributions are narrower or wider than their real counterparts, e.g., the real pole angle in CartPole is box-bounded inside $\pm\,0.418$ rad but the SEs may benefit from not having such a constraint. The synthetic *reward* distribution is wider than the real one, indicating that the sparse reward distribution becomes dense. The green histograms show the SE responses when fed with real environment data based on the logged state-action the agents have seen during testing. Since the green distributions align better with the blue than the orange ones, we conclude it is more likely the shifts are generated by the SE than the agents itself.

Based on these findings, we hypothesize the SEs produce an informed representation of the target environment by modifying the state distributions to bias agents towards relevant states. These results are likely to explain the efficacy of SEs and can be understood as efficient agent "guiding systems". We also observed similar patterns with Dueling DDQN and discrete TD3 (see Appendix A.4).

## 6  EXPERIMENTS WITH REWARD NETWORKS

**Experimental Setup** To better understand our RN experiments, we first describe similarities and dissimilarities to the experimental setup used for SEs. Similar to SEs, to learn RNs, we first identified stable agent and NES HPs (see Appendix B.3) for each of a total of four environments. Contrary to training SEs, we do not vary the agent HPs *during RN training* as we did not observe improvements in doing so. Now, given these HPs, we then trained multiple RNs for 50 NES outer loop iterations with 16 workers for each environment. For CartPole-v0 we used DDQN for RN training and, similar to SEs, evaluated the transferability of DDQN-trained RNs to train Dueling DDQN agents. For the custom environment Cliff Walking (Sutton & Barto, 2018) (see Appendix B.2 for more details) we used Q-Learning (Watkins, 1989) and show transfer to SARSA (Rummery & Niranjan, 1994). We

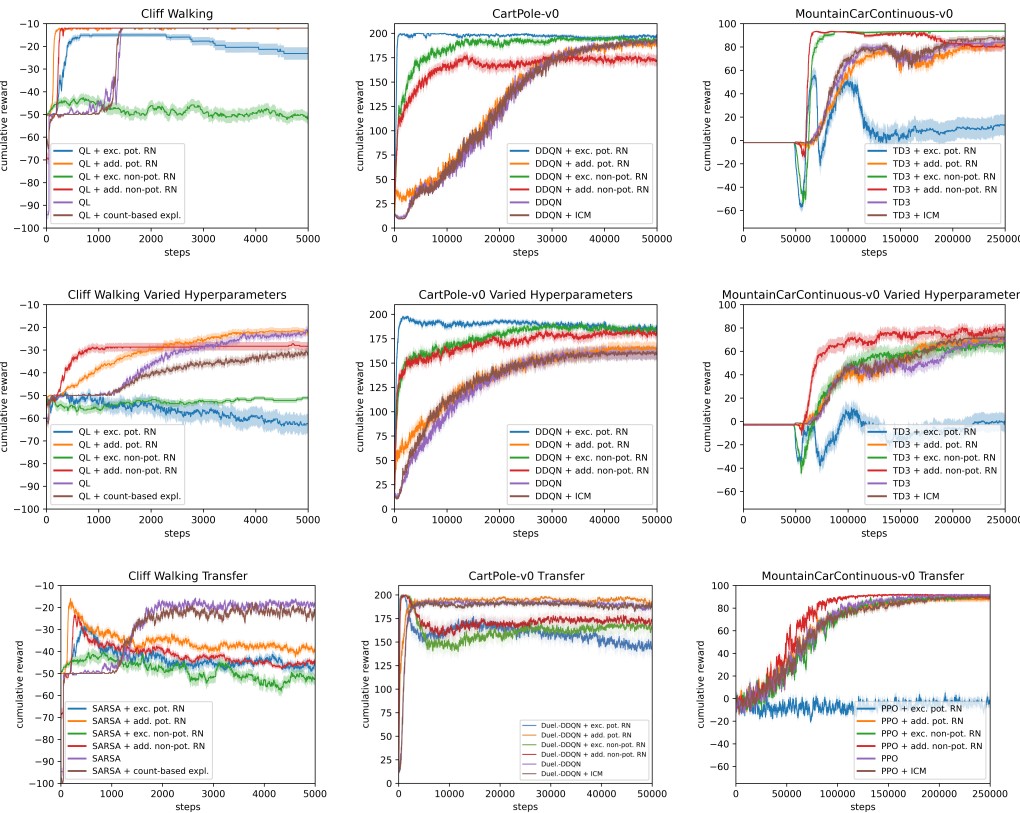

Figure 5: The average cumulative *test* rewards of agents when trained on different RN variants for one episode and evaluated on the real environments for one episode alternatingly. **Top row:** Training and evaluation with the same agent using default agent hyperparameters (HPs). **Center row:** With varied agent HPs. **Bottom row:** Transfer to unseen agents. We show the std. error of the mean and the flat curves in TD3 are due to filling the replay buffer before training.

also include MountainCarContinuous-v0 (Brockman et al., 2016) and HalfCheetah-v3 (Todorov & Tassa, 2012) with TD3 and transfer to PPO (Schulman et al., 2017). With these, we cover a wide range of variation (discrete and continuous state/action spaces, sparse/dense rewards) and computational complexity for training an RN (see Appendix C for details). All RN plots show the average cumulative *test* reward as a function of RN train steps. We alternated episodes of training on the RN and evaluation on the real environment and performed 100 runs (10 RNs × 10 agents) per curve on each environment except for HalfCheetah where we used 25 runs (5 RNs × 5 agents).

**Baselines**  For all environments, we compare our approach to multiple baselines, namely the bare agent and the bare agent extended by an Intrinsic Curiosity Module (ICM) (Pathak et al., 2017) that uses the error of predicted action effects as the synthetic reward. Additionally, for Cliff Walking we use count-based exploration (Strehl & Littman, 2008) with a synthetic reward $\frac{\beta}{n(s,a)}$ to encourage exploration of less visited (s,a)-pairs with $\beta \in \mathbb{R}_{\geq 0}$ and $n : \mathcal{S} \times \mathcal{A} \to \mathbb{N}_0$ a visitation count. For HalfCheetah, we also consider the observation vectors $v$ and $v'$ with the position and velocity of the current and next state that HalfCheetah provides. We use these as input to the RNs with $\Phi(s, v)$ or $\Phi(s', v')$, combine them with the existing variants from Table 1 and tag them with "+ augm.".

**Evaluating the Performance of RN-trained Agents**  Like in the SE case, we study how efficient and hyperparameter-sensitive the RNs are. Consider the top and middle row of Figure 5 that show how the RNs perform in training *seen* agents (used to optimize the RNs) as well as the performance when we vary the agents' HPs (i.e., learning rate, discount factor, etc., see Appendix B.3).

For all environments, we see that there always exists an RN variant that offers significant speed-ups compared to the baselines not involving RNs. The variant *additive potential RN* $(r + \gamma\Phi(s') - \Phi(s))$ implementing the original PBRS formulation (Ng et al., 1999) seems to be a good default choice. However, it often does not outperform the baseline, presumably trading reliability and formal guarantees with performance. The variant *additive non-potential RN* $(r + \Phi(s'))$ seems to often be the better choice, as it never failed completely and often outperformed the original PBRS variant. We argue that *additive non-potential RN* can be seen as an undiscounted *additive potential RN* ($\gamma = 1$ vs. $\gamma = 0.99$) that may be better in sparse-reward and long-trajectory tasks as in ContinuousMountain-Car. In contrast, *exclusive potential RN* $(\gamma\Phi(s') - \Phi(s))$ shows an increased risk of failing and we hypothesize that not involving the real reward may result in a challenging RN optimization. Across all tasks, RNs do not seem to overfit to the HPs (Figure 5, 2nd row). The results and a discussion for HalfCheetah are given in Appendix B.1.

**Transferability of RNs** In this experiment, we investigate the ability of RNs to train unseen agents. For this, consider the bottom row of Figure 5 (Appendix Figure 9 for HalfCheetah). We used the default agent HPs reported in Appendix B.3 without varying them. The results show that RNs can also train unseen RL agents efficiently. However, for both Cliff Walking and CartPole some RN variants cause a deterioration of the learned policies after initially teaching well-performing policies. While the reason for this is unclear, a workaround is to apply early-stopping since *additive potential RN* and *additive non-potential RN* deliver significant (Cliff W., CartPole) to moderate (Mountain-Car) speedups or higher final performance (HalfCheetah) even in the agent-transfer scenario.

We noticed that RNs are generally less prone to overfitting than SEs and are overall easier to train. However, the components of the RN variants seem to induce important inductive biases that beg a deeper understanding and may require a more isolated analysis. Nevertheless, we see clear evidence of the benefits of using RNs for efficient RL agent training. All in all, we argue that the RN efficiency, like for SEs, stems from a learned informed representation that biases agents towards helpful states for completing the task quickly. In the same spirit of the qualitative analysis of SEs in Section 5, a similar study for Cliff Walking RNs that supports this hypothesis is in Appendix B.2.

## 7 LIMITATIONS OF THE APPROACH

As reported in (Salimans et al., 2017), NES methods strongly depend on the number of workers and require a lot of parallel computational resources. We observed this limitation in preliminary SE experiments when we applied our method to more complex environments, such as the HalfCheetah or Humanoid task. Here, 16 workers were insufficient to learn SEs able to solve them. Moreover, non-Markovian states and partial observability may add further complexity to the optimization of SEs and RNs. In contrast to the SE case, in the RN case, scaling our method to more complex tasks was straightforward since learning rewards is intuitively easier than learning complex state dynamics. Nevertheless, the success of learning efficient proxy models also depends significantly on hyperparameter optimization. Lastly, we observed that optimized NES HPs were transferable among different target environments in the RN case but usually not in the SE case.

## 8 CONCLUSION

We proposed a novel method for learning proxy models outside the current learning schemes. We apply it to learn synthetic environments and reward networks for RL environments. When training agents on SEs without involving the real environment at all, our results on two environments show significant reductions in the number of training steps while maintaining the target task performance. In the RN case, we also noticed similar benefits in three of four environments. Our experiments showed that the proxies produce narrower and shifted states as well as dense reward distributions that resemble informed representations to bias agents towards relevant states. We illustrated that these proxies are robust against hyperparameter variation and can transfer to unseen agents, allowing for various applications (e.g., as agent-agnostic, cheap-to-run environments for AutoML, for agent and task analysis or agent pre-training). Overall, we see our method as a basis for future research on more complex tasks and we believe that our results open an exciting new avenue of research.

## ACKNOWLEDGEMENTS

The authors acknowledge funding by Robert Bosch GmbH, by the Deutsche Forschungsgemein-schaft (DFG, German Research Foundation) under grant number 417962828, by the state of Baden-Württemberg through bwHPC and the German Research Foundation (DFG) through grant no INST 39/963-1 FUGG, and by TAILOR, a project funded by the EU Horizon 2020 research and innovation programme under GA No 952215.

## ETHICS STATEMENT

As learned data generating processes that teach learners, our proxies may have ethical and soci-etal consequences. On the application side, we see the risk of learning proxies on maliciously re-purposed environments, for example, that allow efficiently learning policies for weapon systems. Ways to control this may include ethically reviewing both environment releases (staging releases) and the used optimization objectives, as well as regulations to enforce their disclosure. In contrast, proxies can also be used for good-natured applications, such as elderly care robotics to help analyze human safety.

Methodologically, we see risks in generating proxies that are purely optimized for efficiency or effectiveness as they may elicit agent policies that achieve the objective by neglecting established societal and moral rules. Effects like these may be reinforced by the design of the target environ-ment. While the latter could be addressed through the democratized design of target environments, the former could be tackled by using multi-objective optimization during proxy learning and by adopting fair-efficient rewards (Jabbari et al., 2017). The flexibility that the proxy optimization al-lows can also be beneficial to society, for example, by using objectives that enable fuel-saving in (autonomous) vehicles. Moreover, our proxies can have a positive effect on the environment as they target sample inefficiency in RL, e.g., by using them for efficiently pre-training RL agents like in supervised learning and, as addressed in our work, by targeting proxy transferability to exploit in-vested resources across agent families. In summary, despite the possible negative implications of learning environment proxies, we find that there exist possibilities to successfully address them and that the laid out merits outweigh the potential harms.

## REPRODUCIBILITY STATEMENT

In our repository, we include the full code, SE / RN models, hyperparameter configurations, and instructions to reproduce all figures reported in the main paper. All our experiments use random seeds and all results are based on multiple random runs for which standard deviations are reported. Moreover, we dedicate individual experiments to investigate the robustness to variation of randomly sampled hyperparameters in results denoted by *vary HP* in Sections 5 and 6. Additionally, we report the total run-time for SEs and RNs in Sections 5 and 6 and discuss the computational complexity in more detail in Appendix C. Our code can be found here: `https://github.com/automl/learning_environments`

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

## APPENDIX A    SYNTHETIC ENVIRONMENTS

### A.1    HEURISTICS FOR AGENT TRAINING AND EVALUATION

Determining the number of required training episodes $n_e$ on an SE is challenging as the rewards of the SE may not provide information about the current agent's performance on the real environment. Thus, we use a heuristic to early-stop training once the agent's training performance on the *SE* converged. Specifically, let $C_k$ be the cumulative reward of the $k$-th training episode. The two values $\bar{C}_d$ and $\bar{C}_{2d}$ maintain a non-overlapping moving average of the cumulative rewards over the last $d$ and $2d$ episodes up to episode $k$. Now, if $\frac{|\bar{C}_d - \bar{C}_{2d}|}{|\bar{C}_{2d}|} \leq C_{diff}$ the training is stopped. In all SE experiments we choose $d = 10$ and $C_{diff} = 0.01$. Agent training on *real environments* is stopped when the average cumulative reward across the last $d$ test episodes exceeds the solved reward threshold. In case no heuristic is triggered, we train for $n_e = 1000$ episodes at most on SEs and real environments. Independent of whether we train an agent on $\mathcal{E}_{real}$ or $\mathcal{E}_{syn}$, the process to assess the actual agent performance is equivalent: we run the agent on 10 test episodes from $\mathcal{E}_{real}$ for a fixed number of task-specific steps (200 on CartPole and 500 on Acrobot) and use the cumulative rewards as the performance.

### A.2    DISCUSSION ON THE FEASIBILITY OF LEARNING SES WITH OUR ALGORITHM

After identifying stable DDQN and NES hyperparameters (HPs), we ran Algorithm 1 in parallel with 16 workers for 200 NES outer loop iterations for both CartPole and Acrobot. Each worker had one Intel Xeon Gold 6242 CPU core at its disposal, resulting in an overall runtime of ~6-7h on Acrobot and ~5-6h on CartPole for 200 NES outer loop iterations. For reference, we note that (Salimans et al., 2017) used 80 workers each having access to 18 cores for solving the Humanoid task.

In Figure 6 it becomes evident that the proposed method with the given resources and used hyperparameters given by Table 3 allows identifying SEs that can teach agents to solve CartPole and Acrobot tasks respectively. Each thin line in the plot corresponds to the average of 16 worker evaluation scores given by *EvaluateAgent* in Algorithm 1 as a function of the NES outer loop iteration. We repeat this for 40 separate NES optimization runs and visualize the average across the thin lines as a thick line for each task. We note that we only show a random subset of the 40 thin lines for better visualization and randomly sample the seeds at all times in this work. We believe that the stochasticity introduced by this may lead to the variance visible in the plot when searching for good SEs.

Both the stochasticity of natural gradients and the sensitivity of RL agents to seeds and parameter initializations may additionally contribute to this effect. Notice, it is often sufficient to run approximately 50 NES outer loops to find SEs that solve a task. Besides early-stopping, other regularization techniques (e.g. regularizing the SE parameters) can be applied to address overfitting which we likely observe in the advanced training of the Acrobot task.

### A.3    EVALUATION OF PERFORMANCE AND TRANSFERABILITY OF SES ON THE ACROBOT TASK

Similar to Figure 3, we visualize the performance and transferability evaluation results of SEs for the Acrobot-v1 task. Again, the top row depicts the density functions of the average cumulative rewards collected by DDQN, Dueling DDQN and discrete TD3 agents based on 4k evaluations per density and DDQN-trained SEs. Like in the CartPole case, we show three different SE training settings: agents trained on real env. with varying agent HPs (blue), on DDQN-trained SEs when varying agent HPs during NES runs (orange), on DDQN-trained SEs where the agent HPs were fixed during training of the SEs (green). The bottom row visualizes the average reward, train steps, and episodes across the 4k evaluations and each bar correspond to one of the shown densities. We achieve on Dueling DDQN a $\sim 38\%$ (18376 vs. 29531 train steps) and on discrete TD3 a $\sim 82\%$ faster and more stable training compared to the baseline. However, we point out that many of the runs with discrete TD3 also fail (mean reward: -343). This is likely caused by the limited transferability already observed in the Cartpole case but may also due to sub-optimal hyperparameter selection since we reused the DDQN HPs found on Cartpole for Acrobot (as well as the HPs and ranges for

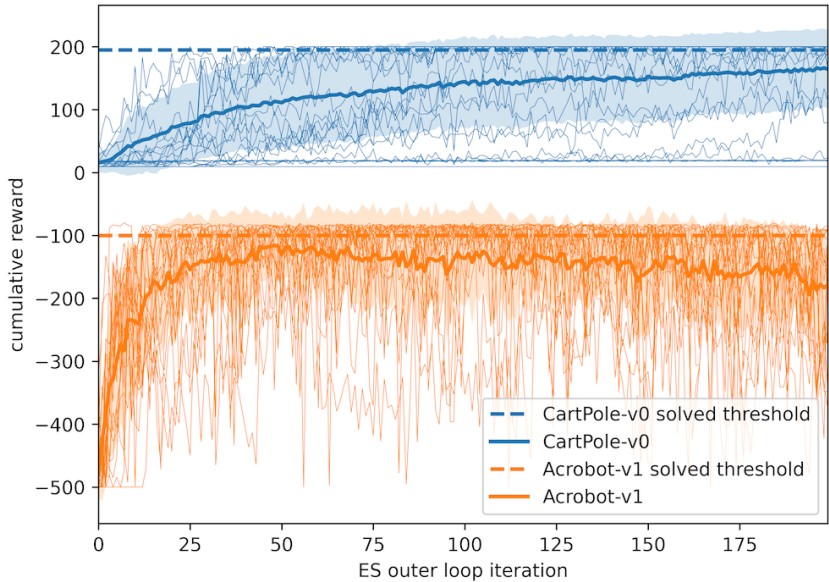

Figure 6: Results from 40 different NES runs with 16 workers each (using random seeds) show that our method is able to identify SEs that allow agents to solve the target tasks. Each thin line corresponds to the average of 16 worker evaluation scores returned by *EvaluateAgent* in our algorithm as a function of the NES outer loop iterations.

variation from Table 2). We further point out that training on SEs consistently yields a more stable training as can be seen in the standard deviations of the train steps and episodes bars.

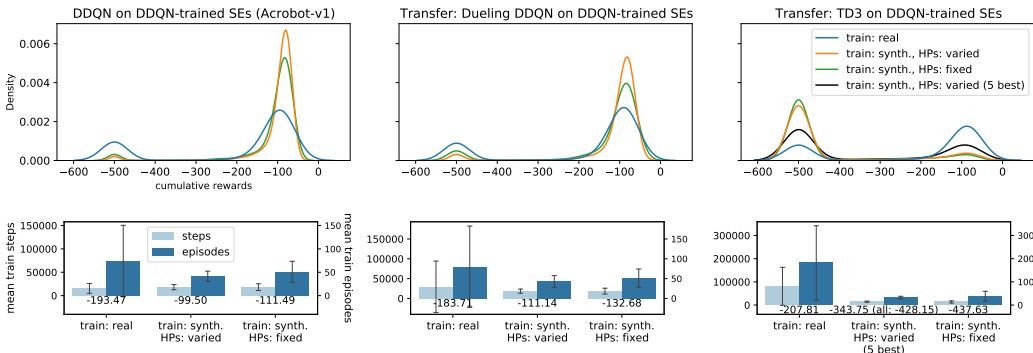

Figure 7: Evaluation of performance and transferability of SEs on the Acrobot-v1 task.

## A.4 QUALITATIVE STUDY OF SYNTHETIC ENVIRONMENTS FOR THE CARTPOLE AND ACROBOT TASKS

To extend the qualitative analysis of SE behavior from Section 9, we depict in Figure 8 additional histograms for the approximate next state $s'$ and reward $r$ distributions produced by 10 agents with random seeds and default hyperparameters from Table 5 for each histogram plot. We here additionally show the behavior for Acrobot and also for Dueling DDQN and discrete TD3 agents. For the histograms with discrete TD3 we chose well-performing but arbitrary SE model according the individual model analysis, see Section F.1.1 and F.1.2). We chose a well-performing model in the case of discrete TD3 since we assumed that the significant number of failing runs would deteriorate the discrete TD3 performance and distort the comparison to the other (well-performing) DDQN and Dueling DDQN agents. For the DDQN and Dueling DDQN agents, we chose the SE model completely at random, independent of their performance. In summary, we see the same patterns that we

observed for CartPole and DDQN in Section 9 also with Dueling DDQN and discrete TD3 on both tasks: distribution shift, narrower synthetic distributions (and state-stabilizing or control behavior, meaning that the number of large positive and negative states are rarely encountered, for example, this can be seen in the cart velocity (CartPole) and the joint angular velocity (Acrobot) plots), and wider/dense reward distributions. The histograms in green show the SE responses when fed with real environment data based on the logged state-action the agents have seen during testing. Since the green distributions align better with the blue than the orange ones, we conclude it is more likely the shifts are generated by the SE than the agent. For CartPole we show all state dimensions and for Acrobot we show four of the six state dimensions due to brevity.

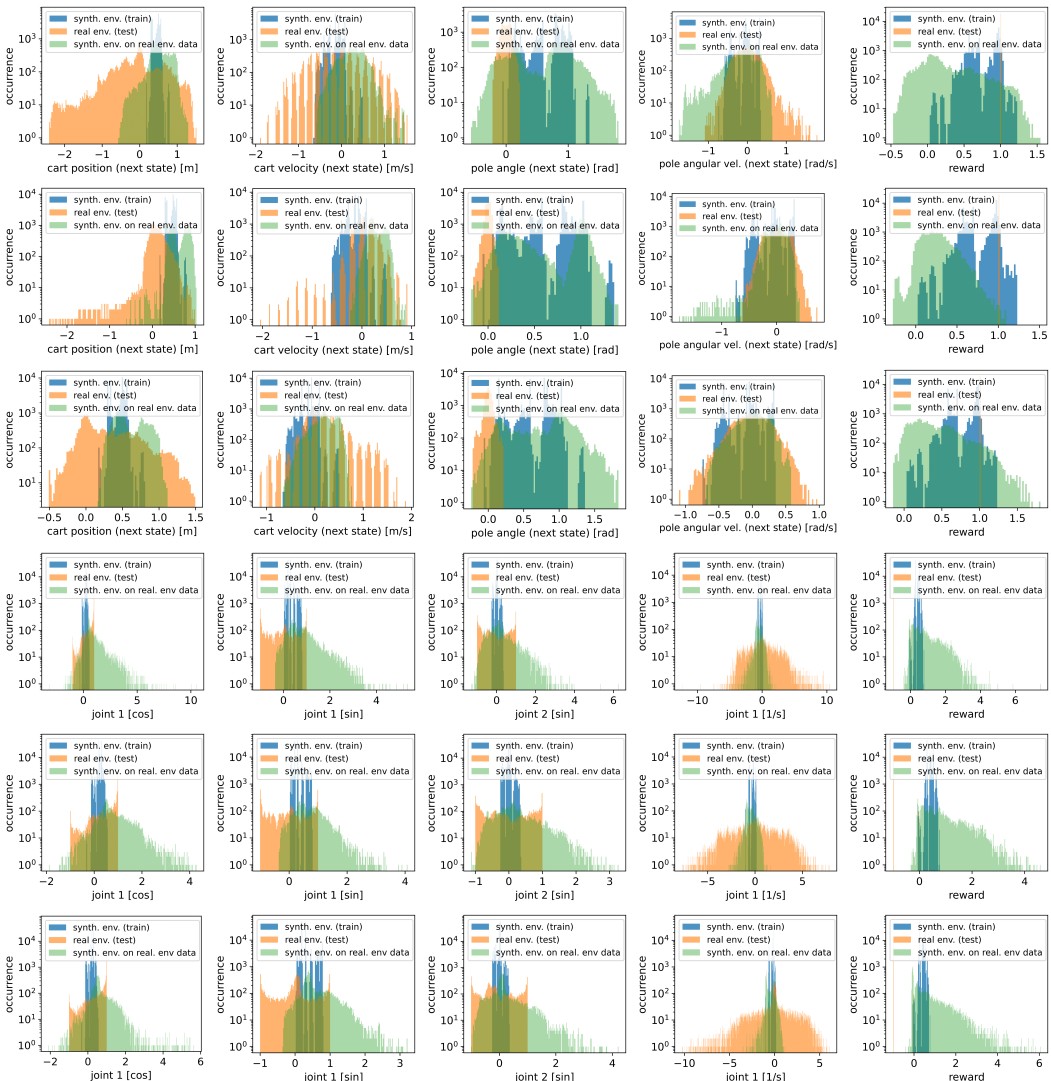

Figure 8: Histograms of approximate next state $s'$ and reward $r$ distributions produced by 10 DDQN or Dueling DDQN agents when trained on an SE (blue) and when afterwards tested for 10 episodes on a real environment (orange) for each task. **1st row**: CartPole and DDQN, **2nd row**: CartPole and Dueling DDQN, **3rd row**: CartPole and discrete TD3, **4th row**: Acrobot and DDQN, **5th row**: Acrobot and Dueling DDQN, **6th row**: Acrobot and discrete TD3

The average cumulative rewards across the 10 runs for each agent variant and task are:

- CartPole and DDQN (1st row): 199.3

- CartPole and Dueling DDQN (2nd row): 193.75

- CartPole and discrete TD3 (3rd row): 197.26

- Acrobot and DDQN (4th row): -91.62
- Acrobot and Dueling DDQN (5th row): -88.56
- Acrobot and discrete TD3 (6th row): -93.17

## A.5 Synthetic Environment Hyperparameters

The following table provides an overview of the agent and NES hyperparameter (HP) ranges that we optimized in an additional outer loop of our algorithm. The HPs in the 2nd and 3rd columns represent the results of this optimization and which we used for learning SEs. Note that in experiments where we sampled agent HPs from ranges given by Table 2 we overwrite a small subset (namely DDQN learning rate, batch size, DDQN no. of hidden layers, and DDQN hidden layer size) of the listed HPs below (runs denoted by "HPs: varied").

Table 3: Hyperparameter configuration spaces optimized for learning SEs.

| Hyperparameter | CartPole-v0 | Acrobot-v1 | Optimization value range | log. scale |
|---|---|---|---|---|
| NES step size ($\alpha$) | 0.148 | 0.727 | $0.1 - 1$ | True |
| NES noise std. dev. ($\sigma$) | 0.0124 | 0.0114 | $0.01 - 1$ | True |
| NES mirrored sampling | True | True | False/True | - |
| NES score transformation | All Better 2 | All Better 2 | see Appendix Section E.1 | - |
| NES SE no. of hidden layers | 1 | 1 | $1 - 2$ | False |
| NES SE hidden layer size | 83 | 167 | $48 - 192$ | True |
| NES SE activation function | LReLU | PReLU | Tanh/ReLU/LReLU/PReLU | - |
| DDQN initial episodes | 1 | 20 | $1 - 20$ | True |
| DDQN batch size | 199 | 149 | $64 - 256$ | False |
| DDQN learning rate | 0.000304 | 0.00222 | $0.0001 - 0.005$ | True |
| DDQN target net update rate | 0.00848 | 0.0209 | $0.005 - 0.05$ | True |
| DDQN discount factor | 0.988 | 0.991 | $0.9 - 0.999$ | True (inv.) |
| DDQN initial epsilon | 0.809 | 0.904 | $0.8 - 1$ | True |
| DDQN minimal epsilon | 0.0371 | 0.0471 | $0.005 - 0.05$ | True |
| DDQN epsilon decay factor | 0.961 | 0.899 | $0.8 - 0.99$ | True (inv.) |
| DDQN no. of hidden layers | 1 | 1 | $1 - 2$ | False |
| DDQN hidden layer size | 57 | 112 | $48 - 192$ | True |
| DDQN activation function | Tanh | LReLU | Tanh/ReLU/LReLU/PRelu | - |

The following table lists the hyperparameters that we always kept fixed and which were never optimized:

Table 4: Subset of the hyperparameter configuration spaces that we kept fixed (not optimized) for learning SEs.

| Hyperparameter | Symbol | CartPole-v0 | Acrobot-v1 |
|---|---|---|---|
| NES number of outer loops | $n_o$ | 200 | 200 |
| NES max. number of train episodes | $n_e$ | 1000 | 1000 |
| NES number of test episodes | $n_{te}$ | 10 | 10 |
| NES population size | $n_p$ | 16 | 16 |
| DDQN replay buffer size | - | 100000 | 100000 |
| DDQN early out number | $d$ | 10 | 10 |
| DDQN early out difference | $C_{diff}$ | 0.01 | 0.01 |
| env. max. episode length | - | 200 | 500 |
| env. solved reward | - | 195 | $-100$ |

The table below lists the agent hyperparameters (HPs) that we chose as default. We used these HPs in all experiments after SEs were found, i.e. for evaluations in Figures 3, 4, and 7. Note that in some cases we sampled agent HPs from ranges given by Table 2. In these cases (runs denoted by "HPs: varied", "HPs: fixed", and also the individual SE models plots below) the HPs of the table are overwritten by the sampled ones.

Table 5: Default agent hyperparameters for the evaluations depicted in Figure 3, 4, and 7. *DDQN early out number* and *DDQN early out difference* are equivalent to Table 4.

| Agent hyperparameter | Default value |
|---|---|
| initial episodes | 10 |
| batch size | 128 |
| learning rate (DDQN & D.DDQN / TD3) | 0.001 / 0.0005 |
| target network update rate | 0.01 |
| discount factor | 0.99 |
| epsilon decay factor | 0.9 |
| number of hidden layers | 2 |
| hidden layer size | 128 |
| activation function (DDQN & D. DDQN / TD3) | ReLU / Tanh |
| replay buffer size | 100000 |
| max. train episodes | 1000 |
| Gumbel Softmax start temperature / one-hot-encoded actions (TD3) | 1 / False |

## APPENDIX B    REWARD NETWORKS

### B.1    HALFCHEETAH RNS PLOTS

For HalfCheetah, we see that *additive non-potential RN* (and also *exclusive non-potential RN*) trains well-performing agents the fastest but saturates at a cumulative reward of ∼3k which corresponds to the used solved reward threshold. This may be due to the used objective function since its term $max(0, C_{sol} - C_{fin})$ does not incentivize RNs to improve after reaching the threshold.

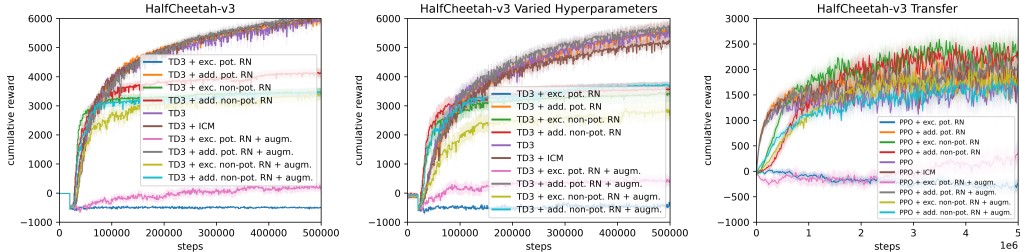

Figure 9: Performance of different RN variants for the HalfCheetah-v3 environment. Left: performance of RNs that train the known TD3 agent with default hyperparameters. Center: performance of RNs when the hyperparameters of known agents are varied. Right: Transfer performance to the unseen PPO agents with default hyperparameters. All curves denote the mean cumulative reward across 25 runs (5 agents × 5 RNs). Shaded areas show the standard error of the mean and the flat curves in TD3 stem from a fixed number of episodes that are used to fill the replay buffer before training. Notice that using the observation vector $v$ in the RN variants (denoted by "+augm.") only marginally yields a higher train efficiency.

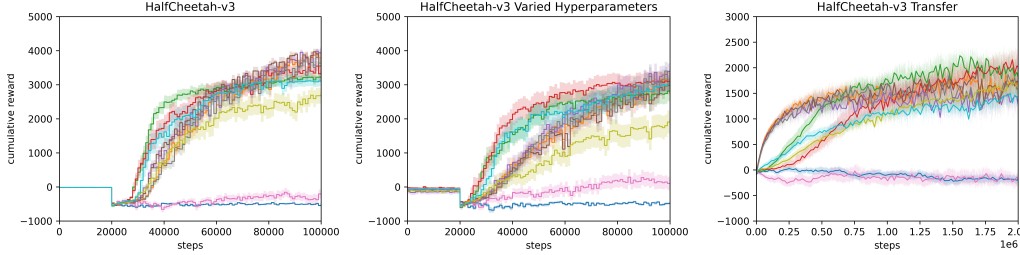

Figure 10: This is the zoomed in version of Figure 9.

### B.2    QUALITATIVE STUDY OF REWARD NETWORKS ON THE CLIFF WALKING TASK

Like in Section A.4, we again make use of the small state and action space and analyze the learned augmented rewards of the proposed different RN variants across 50 generated RN models on our Cliff Walking environment (Sutton et al., 2020). Cliff Walking consists of a two-dimensional grid with 4×12 states in which the agent, starting at the start state (S), has to reach the goal state (G) by moving up, down, left, or right to adjacent states. 10 of the states in the lower row are cliffs. The agent receives a reward of −1 for every action it takes and a reward of −100 if it enters a cliff state. The episode terminates if the agent enters either the goal state or a cliff state. The environment is further bounded by walls, e.g. if the agent moves left when being in the state (S), it stays in the state (S).

In Figure 11 we depict a visual representation of the learned rewards for different RN variants averaged across 50 RN models taken from the RN experiment in Section 6 (i.e. trained with a Q-Learning agent with non-varying HPs, no early stopping criterion in the outer loop, 50 outer loop iterations). To do this, we color-coded the rewards we received from the RNs for all possible state-action pairs. To visualize the rewards corresponding to each of the four actions, every state is split up into four triangles corresponding to the different actions (move up, down, left, right) that the agent can perform when being in this state. We show small, average, and high rewards in red, yellow, and green, respectively. The left column of Figure 11 shows the grids with all rewards

where we can observe that the negative rewards of $-100$ of the additive RNs distort the color plot such that the fine-grained reward structure learned by the RNs becomes invisible. To address this, the right column features the same plots while ignoring any rewards $\leq -50$. This implies for the additive RN variants, that the rewards in the cliff states are ignored (white color) since the additive RN variants produce large negative rewards and lead to the described distortion of fine-grained rewards. For the goal state we further note that, although actions were never executed in this terminal state, the (neural network of the) RNs still compute rewards in this case which we did not ignore in the plot simply because it did not lead to distortions.

Matching the performances shown in Figure 5 (top row, left), the reward structure learned by the exclusive potential RN is sub-optimal as the high rewards are nearly randomly distributed. In contrast, the additive potential RNs clearly assign low rewards to any state-action pair that lets the agent enter a cliff state. The RNs further output a low reward for transitions that go from the third row up to the second row and a high reward for the contrarily directed transitions. This keeps the agents close to the cliff and reduces the exploration space which results in faster training. The exclusive non-potential RNs assign useful rewards around the start and goal states but fail to adequately shape the reward along the shortest path from the start to the goal state. Lastly, the additive non-potential RNs clearly assign high rewards that guide the agent along the shortest path towards the goal state while avoiding cliff states. As expected, the well-performing RN variants seem to learn informed reward representations of the task that guide the agent towards relevant states for completing the task like in the SE case.

### B.3 REWARD NETWORK HYPERPARAMETERS

#### B.3.1 GENERAL NES AND ENVIRONMENT HYPERPARAMETERS

The below table lists all HPs for learning RNs that we kept fixed (not optimized) and which are identical across all RN experiments:

Table 6: Hyperparameters that we kept fixed (not optimized) for learning RNs.

| Hyperparameter | Symbol | Value |
|---|---|---|
| NES number of outer loops | $n_o$ | 50 |
| NES number of test episodes | $n_{te}$ | 1 |
| NES population size | $n_p$ | 16 |
| NES unsolved weight | $w_{sol}$ | 100 |

Below table lists all HPs for learning RNs that we optimized and which are identical across all experiments. The NES score transformations are described in detail in Appendix Section E.1 and we also report the results for the BOHB hyperparameter optimization of different score transformation variants in Appendix Section E.2.

Table 7: Hyperparameters that we optimized and which are identical across all RN experiments

| Hyperparameter | Symbol | Optimized value |
|---|---|---|
| NES mirrored sampling | - | True |
| NES score transformation | - | All Better 2 (see Appendix Section E.1) |
| NES step size | $\alpha$ | 0.5 |
| NES noise std. dev. | $\sigma_G$ | 0.1 |

Below table lists the environment-specific HPs:

Table 8: Environment-specific hyperparameters

| Hyperparameter | Cliff Walking | CartPole-v0 | MountainCarContinuous-v0 | HalfCheetah-v3 |
|---|---|---|---|---|
| max. episode length | 50 | 200 | 999 | 1000 |
| solved reward | -20 | 195 | 90 | 3000 |
| NES max. number of train episodes | 100 | 100 | 100 (TD3) 1000 (PPO) | 100 (TD3) 1000 (PPO) |

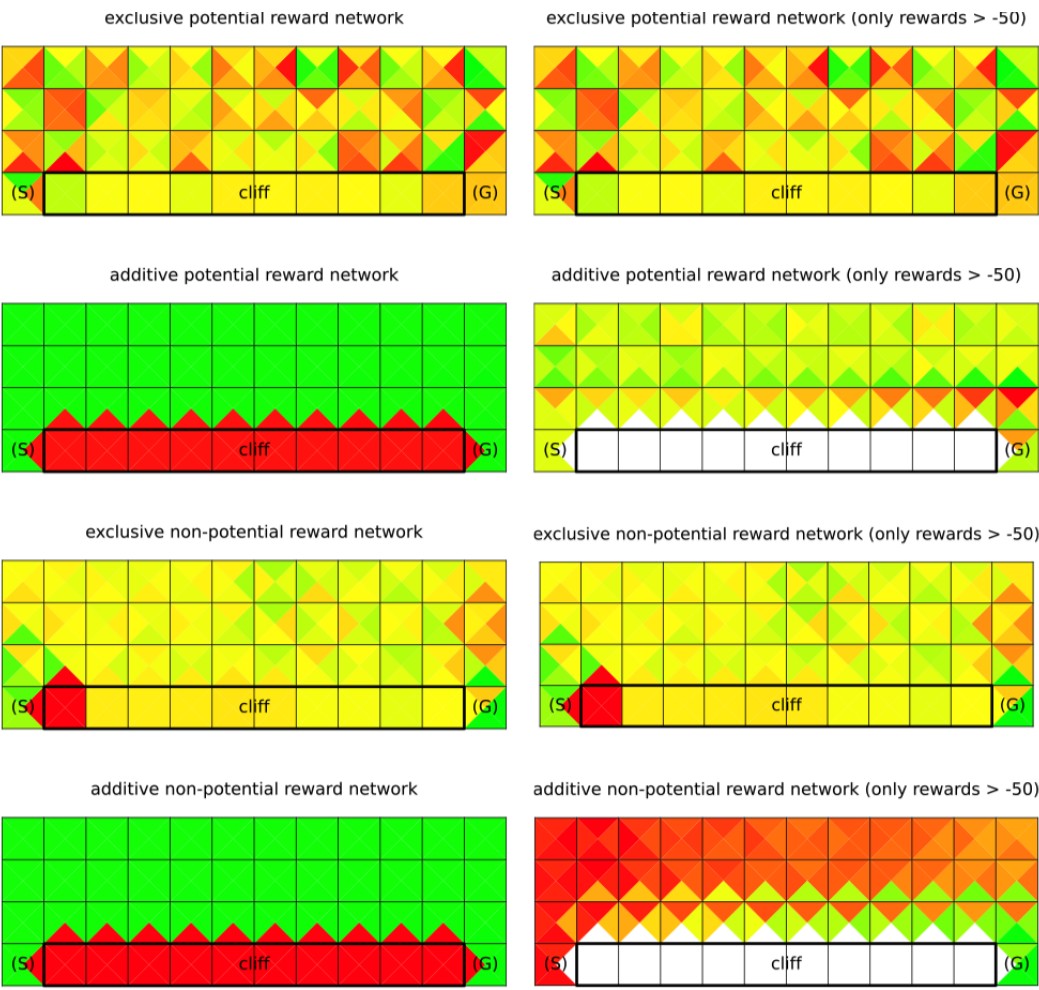

Figure 11: Learned Cliff Walking reward for different reward network types. Shown is the start state (S), goal state (G), all cliff states in the lower row and the learned reward of each state-action pair (red: low reward, yellow: medium reward, green: high reward). The plots on the left side show averaged values across 50 reward networks when considering all rewards, the plots on the right side ignore all rewards that are $\leq -50$. Values have been normalized to be in a [0,1] range in both cases (left with full reward range, right without ignored rewards range).

### B.3.2 VARIED AGENT HYPERPARAMETERS

The following tables list the for Q-Learning, DDQN and TD3 agent hyperparameter ranges from which we sampled in the HP variation experiment.

Table 9: Hyperparameter sampling ranges for the reward networks HP variation experiment

| QL hyperparameter | Value Range | log. scale |
|---|---|---|
| learning rate | 0.1 - 1 | False |
| discount factor | 0.1 - 1 | False |

| DDQN hyperparameter | Value range | log. scale |
|---|---|---|
| learning rate | $8.3 \cdot 10^{-5}$ - $7.5 \cdot 10^{-4}$ | True |
| batch size | 11 - 96 | True |
| hidden size | 21 - 192 | True |
| hidden layer | 1 - 2 | False |

| TD3 hyperparameter | Value range | log. scale |
|---|---|---|
| learning rate | $10^{-4}$ - $9 \cdot 10^{-4}$ | True |
| batch size | 85 - 768 | True |
| hidden size | 42 - 384 | True |
| hidden layer | 1 - 3 | False |

### B.3.3 AGENT AND NES HYPERPARAMETERS FOR CLIFF WALKING

In Table 10 we list the NES and Q-Learning hyperparameter configuration spaces that we optimized for learning RNs on Cliff Walking. In Table 11 we list the default Q-Learning and SARSA (transfer) hyperparameters that we used once RNs have been learned. In Table 12 we list the hyperparameter configuration spaces that we used (and optimized over) for the count-based Q-Learning baseline.

Table 10: Hyperparameter configuration spaces optimized for learning RNs on Cliff Walking.

| Hyperparameter | Optimized value | Value range | log. scale |
|---|---|---|---|
| NES RN number of hidden layers | 1 | $1 - 2$ | False |
| NES RN hidden layer size | 32 | $32 - 192$ | True |
| NES RN activation function | PReLU | Tanh/ReLU/LReLU/PReLU | - |
| Q-Learning learning rate | 1 | $0.001 - 1$ | True |
| Q-Learning discount factor | 0.8 | $0.001 - 1$ | True (inv.) |
| Q-Learning initial epsilon | 0.01 | $0.01 - 1$ | True |
| Q-Learning minimal epsilon | 0.01 | $0.01 - 1$ | True |
| Q-Learning epsilon decay factor | 0.0 | $0.001 - 1$ | True (inv.) |

Table 11: Default Q-Learning and SARSA hyperparameters that we used after learning RNs.

| Hyperparameter | Default value |
|---|---|
| Q-Learning & SARSA learning rate | 1 |
| Q-Learning & SARSA discount factor | 0.8 |
| Q-Learning & SARSA initial epsilon | 0.1 |
| Q-Learning & SARSA minimal epsilon | 0.1 |
| Q-Learning & SARSA epsilon decay factor | 0.0 |

Table 12: Used baseline hyperparameter configuration spaces

| Hyperparameter | Optimized value | Value range | log. scale |
|---|---|---|---|
| Count-based Q-Learning $\beta$ | 0.1 | $0.0001 - 2$ | False |

### B.3.4 AGENT AND NES HYPERPARAMETERS FOR CARTPOLE-V0

In Table 13 we list the NES and DDQN hyperparameter configuration spaces that we optimized for learning RNs on CartPole. We do not list the optimization value range in this table since they are equivalent as in the learning SE case described in Table 3. In Table 14 we list the default DDQN and Dueling DDQN (transfer) hyperparameters that we used once RNs have been learned. In Table 13 we list the hyperparameter configuration spaces that we used (and optimized over) for the ICM baseline (Pathak et al., 2017). We note that, despite optimizing the ICM HPs, we did not find that the optimized ICM HPs yielded better results than the default ones which is why we use the default ICM HPs in all experiments that involve ICM.

Table 13: Optimized hyperparameters for learning RNs on CartPole. The value ranges are the same as in Table 3. Hyperparameters not listed were not optimized.

| Hyperparameter | Optimized value |
|---|---|
| NES RN number of hidden layers | 1 |
| NES RN hidden layer size | 64 |
| NES RN activation function | PReLU |
| DDQN initial episodes | 1 |
| DDQN batch size | 192 |
| DDQN learning rate | 0.003 |
| DDQN target network update rate | 0.01 |
| DDQN discount factor | 0.99 |
| DDQN initial epsilon | 0.8 |
| DDQN minimal epsilon | 0.03 |
| DDQN epsilon decay factor | 0.95 |
| DDQN number of hidden layers | 1 |
| DDQN hidden layer size | 64 |
| DDQN activation function | LReLU |
| DDQN replay buffer size | 1000000 |

Table 14: Default DDQN and Dueling DDQN hyperparameters that we used after learning RNs.

| Hyperparameter | Default value |
|---|---|
| DDQN & Dueling DDQN initial episodes | 1 |
| DDQN & Dueling DDQN batch size | 32 |
| DDQN & Dueling DDQN learning rate | 0.00025 |
| DDQN & Dueling DDQN target network update rate | 0.01 |
| DDQN & Dueling DDQN discount factor | 0.99 |
| DDQN & Dueling DDQN initial epsilon | 1.0 |
| DDQN & Dueling DDQN minimal epsilon | 0.1 |
| DDQN & Dueling DDQN epsilon decay factor | 0.9 |
| DDQN & Dueling DDQN number of hidden layers | 1 |
| DDQN & Dueling DDQN hidden layer size | 64 |
| DDQN & Dueling DDQN activation function | ReLU |
| DDQN & Dueling DDQN replay buffer size | 1000000 |
| Dueling DDQN feature dimension | 128 |
| Dueling DDQN value and adv. stream no. hidden layers | 1 |
| Dueling DDQN value and adv. stream hidden layer size | 128 |

### B.3.5 AGENT AND NES HYPERPARAMETERS FOR MOUNTAINCARCONTINUOUS-V0 AND HALFCHEETAH-V3

In Table 16 we list the NES and TD3 hyperparameter configuration spaces that we optimized for learning RNs on MountainCarContinuous (CMC) and HalfCheetah (HC). In Table 17 we list the default TD3 hyperparameters that we used once RNs have been learned. Table 18 shows the PPO agent hyperparameters we used for showing transfer of RNs to train PPO agents. In Table 19 we list the hyperparameter configuration spaces that we used (and optimized over) for the ICM baseline (Pathak et al., 2017). We note that, despite optimizing the ICM HPs, we did not find that the optimized ICM HPs yielded better results than the default ones which is why we use the default ICM HPs in all experiments that involve ICM.

Table 15: Used ICM (Pathak et al., 2017) baseline hyperparameter configuration spaces optimized with CartPole and DDQN. We additionally report the default values taken from the ICM reference implementation.

| Hyperparameter | Default value | Optimized value | Value range | log. scale |
|---|---|---|---|---|
| ICM learning rate | 0.0001 | 0.00001 | $0.00001 - 0.001$ | True |
| ICM $\beta$ | 0.2 | 0.05 | $0.001 - 1$ | True |
| ICM $\eta$ | 0.5 | 0.03 | $0.001 - 1$ | True |
| ICM feature dimension | 64 | 32 | 16-256 | True |
| ICM hidden layer size (forward, feature and inverse model) | 128 | 128 | 16-256 | True |
| ICM number of hidden layers (forward, feature and inverse model) | 2 | - | not optimized | - |
| ICM activation function | LReLU | - | not optimized | - |
| ICM number of residual blocks | 4 | - | not optimized | - |

Table 16: Optimized hyperparameters for learning RNs on MountainCarContinuous-v0 (short "CMC") and HalfCheetah-v3 (short "HC"). Hyperparameters not listed were not optimized.

| Hyperparameter | Optimized value | Value range | log. scale |
|---|---|---|---|
| NES RN number of hidden layers | 1 | 1-2 | False |
| NES RN hidden layer size | 128 | 48-192 | True |
| NES RN activation function | PReLU | Tanh/ReLU/LReLU/PReLU | - |
| TD3 initial episodes (CMC/HC) | 50 / 20 | 1-50 | True |
| TD3 batch size | 192 | 64 - 256 | False |
| TD3 learning rate | 0.003 | 0.0001 - 0.005 | True |
| TD3 target network update rate | 0.01 | 0.005 - 0.05 | True |
| TD3 discount factor (CMC/HC) | 0.99 / 0.98 | 0.9 - 0.99 | True |
| TD3 policy update delay factor | 1 | 1-3 | False |
| TD3 number of action repeats (CMC/HC) | 2 / 1 | 1-3 | False |
| TD3 action noise std. dev. | 0.05 | 0.05 - 0.2 | True |
| TD3 policy noise std. dev. | 0.2 | 0.1 - 0.4 | True |
| TD3 policy noise std. dev. clipping | 0.5 | 0.25 - 1 | True |
| TD3 number of hidden layers | 2 | 1-2 | False |
| TD3 hidden layer size | 128 | 48 - 192 | True |
| TD3 activation function (CMC/HC) | LReLU / ReLU | Tanh/ReLU/LReLU/PReLU | - |
| TD3 replay buffer size | 100000 | not optmized | - |

Table 17: Default TD3 hyperparameters that we used after learning RNs on MountainCarContinuous-v0 (short "CMC") and HalfCheetah-v3 (short "HC").

| Hyperparameter | Default value |
|---|---|
| TD3 initial episodes (CMC/HC) | 50 / 20 |
| TD3 batch size | 256 |
| TD3 learning rate | 0.0003 |
| TD3 target network update rate | 0.005 |
| TD3 discount factor | 0.99 |
| TD3 policy update delay factor | 2 |
| TD3 number of action repeats (CMC/HC) | 2 / 1 |
| TD3 action noise std. dev. | 0.1 |
| TD3 policy noise std. dev. | 0.2 |
| TD3 policy noise std. dev. clipping | 0.5 |
| TD3 number of hidden layers | 2 |
| TD3 hidden layer size | 128 |
| TD3 activation function | ReLU |
| TD3 replay buffer size | 100000 |

Table 18: PPO (clip) hyperparameters that we used for showing the transfer of RNs on MountainCarContinuous-v0 (short "CMC") and HalfCheetah-v3 (short "HC").

| Hyperparameter | Default value |
|---|---|
| PPO epochs (CMC/HC) | 80 / 10 |
| PPO learning rate | 0.0003 / 0.00001 |
| PPO number of episode updates (CMC/HC) | 10 / 1 |
| PPO discount factor | 0.99 |
| PPO policy update delay factor | 2 |
| PPO number of action repeats (CMC/HC) | 5 / 1 |
| PPO value function coefficient | 1 |
| PPO entropy coefficient (CMC/HC) | 0.01 / 0.001 |
| PPO clip ratio | 0.2 |
| PPO number of hidden layers | 2 |
| PPO hidden layer size (CMC/HC) | 64 / 128 |
| PPO activation function (CMC/HC) | ReLU / tanh |

Table 19: Used ICM (Pathak et al., 2017) baseline hyperparameter configuration spaces optimized on MountainCarContinuous-v0 (short "CMC") and HalfCheetah-v3 (short "HC") with TD3. We additionally report the default values taken from the ICM reference implementation.

| Hyperparameter | Default value | Optimized value | Value range | log scale |
|---|---|---|---|---|
| ICM learning rate (CMC/HC) | 0.0001 | 0.0005 / 0.00001 | $0.00001 - 0.001$ | True |
| ICM $\beta$ (CMC/HC) | 0.2 | 0.1 / 0.001 | $0.001 - 1$ | True |
| ICM $\eta$ (CMC/HC) | 0.5 | 0.01 / 0.1 | $0.001 - 1$ | True |
| ICM feature dimension | 64 | 32 | 16-256 | True |
| ICM hidden layer size (forward, feature and inverse model) | 128 | 128 | 16-256 | True |
| ICM number of hidden layers (forward, feature and inverse model) | 2 | - | not optimized | - |
| ICM activation function | LReLU | - | not optimized | - |
| ICM number of residual blocks | 4 | - | not optimized | - |

### B.4 STUDYING THE INFLUENCE OF DIFFERENT REWARD NETWORK OPTIMIZATION OBJECTIVES

After we noticed the efficiency improvements of SEs, we tested whether RNs could also yield such improvements by directly optimizing for efficiency. Another reason for optimizing for efficiency directly was that the real environment is involved in the RN case and we pursued the goal to reduce real environment interactions. Based on this, we developed the RN objective reported in the paper in Section 4 which we refer to as the *Reward Threshold* objective. However, to analyse whether different objectives would yield similar results, we conducted a small study on the environments to assess the influence of different objectives.

For this, we considered the area-under-the-curve of the cumulative returns objective (*AUC*) as well as the cumulative reward maximization objective (*max reward*) which is the reward threshold objective with just the $C_{fin}$ term.

The overall conclusion from these results is that an AUC objective generally yields similar results as for the reward threshold objective, but in some cases it seems to minorly reduce the efficiency. However, the overall significant efficiency gains remain. Moreover, in contrast to SEs, efficiency improvements in the RN case seem to arise more often when explicitly optimized for it (compare AUC vs. reward threshold below) while maximizing reward (only tested for CartPole and Cliff environments) leads to efficiency improvements only in the CartPole but not in the Cliff environment.

#### B.4.1 CLIFF

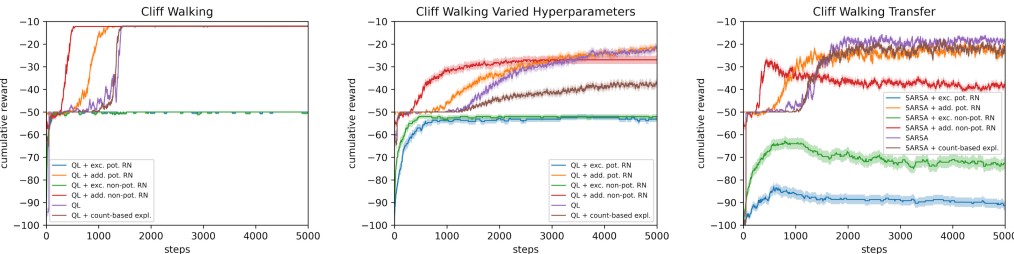

Figure 12: The Cliff environment RNs optimized with the AUC objective.

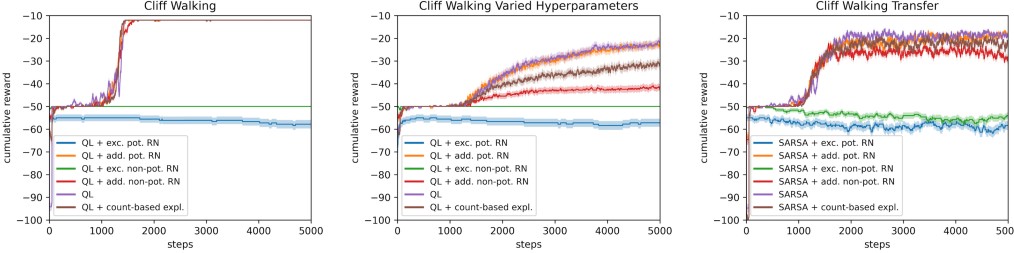

Figure 13: The Cliff environment RNs optimized with the max reward objective.

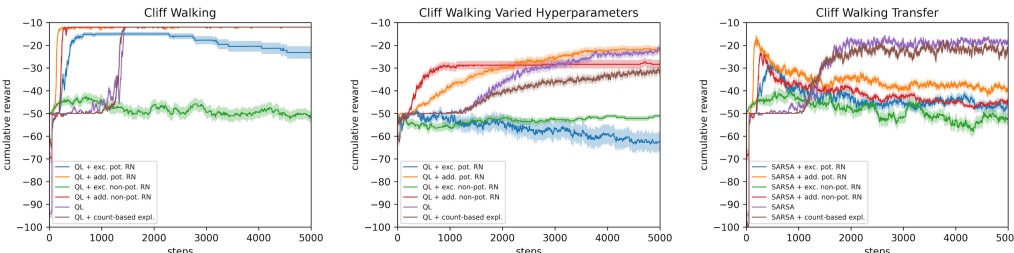

Figure 14: The Cliff environment RNs optimized with the reward threshold objective (similar to the results reported in Section 6).

### B.4.2 CARTPOLE

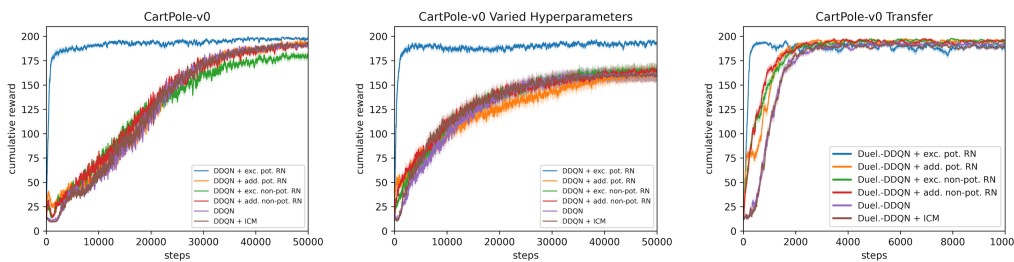

Figure 15: The CartPole environment RNs optimized with the AUC objective.

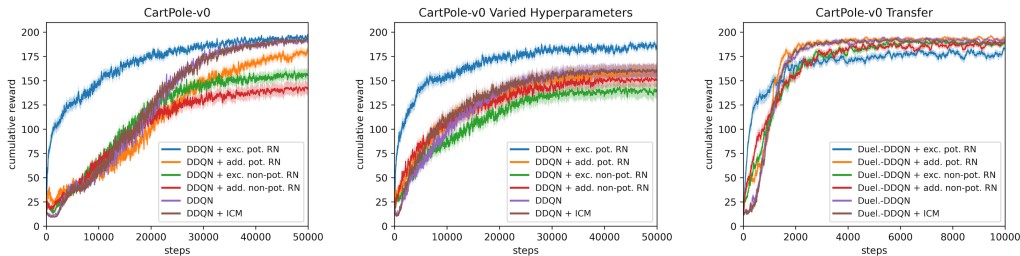

Figure 16: The CartPole environment RNs optimized with the max reward objective.

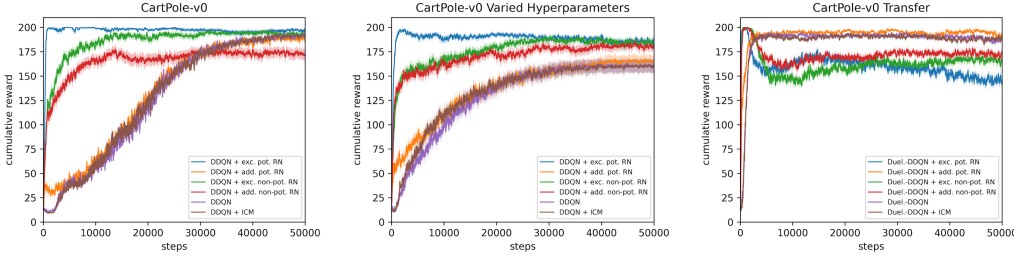

Figure 17: The CartPole environment RNs optimized with the reward threshold objective (similar to the results reported in Section 6).

### B.4.3 MOUNTAINCARCONTINUOUS

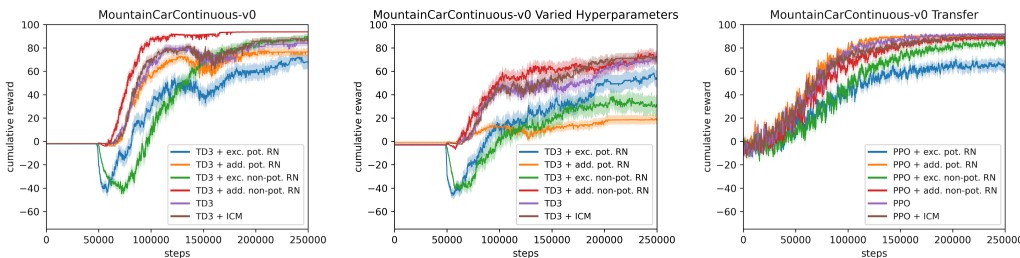

Figure 18: The MountainCarContinuous environment RNs optimized with the AUC objective.

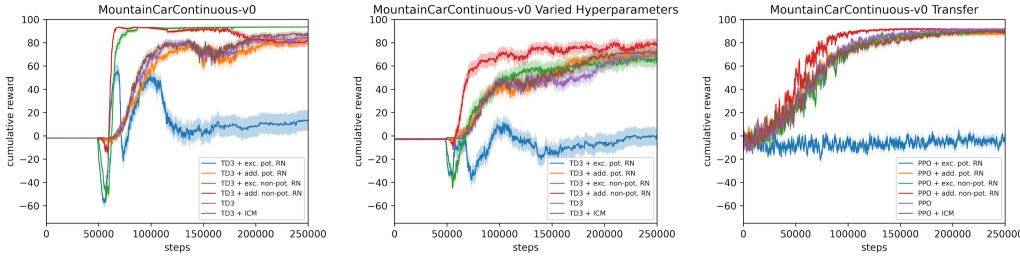

Figure 19: The MountainCarContinuous environment RNs optimized with the reward threshold objective (similar to the results reported in Section 6).

### B.5 HALFCHEETAH

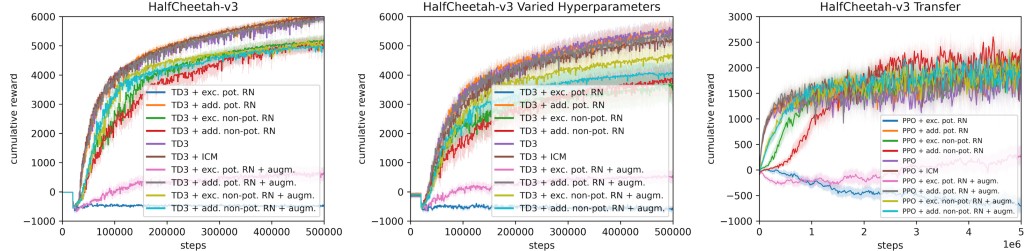

Figure 20: The HalfCheetah environment RNs optimized with the AUC objective.

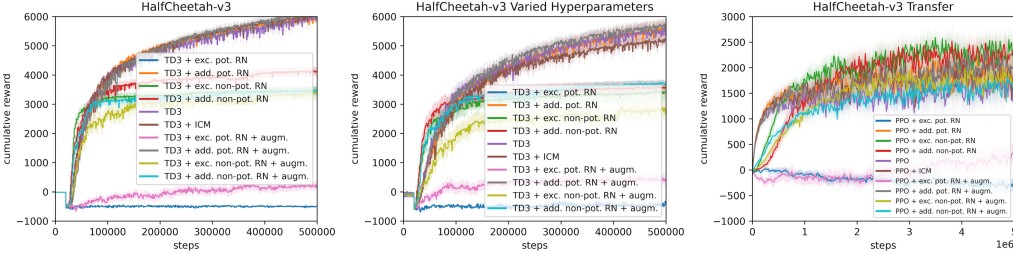

Figure 21: The HalfCheetah environment RNs optimized with the reward threshold objective (similar to the results reported in Appendix Section B.1).

## APPENDIX C   COMPUTATIONAL COMPLEXITY FOR TRAINING SYNTHETIC ENVIRONMENTS AND REWARD NETWORKS

Training both SEs and RNs can be computationally demanding, depending on the environment complexity and the available resources per worker. Table 20 lists approximate timings for a single NES outer loop iteration based on our log files. Every population member had one Intel Xeon Gold 6242 CPU core at its disposal.

Table 20: Approximate timings for one NES outer loop for SEs and RNs

| Environment | Synthetic Environment | Reward Network |
|---|---|---|
| Cliff Walking | - | 10s |
| Acrobot-v1 | 100s | - |
| CartPole-v0 | 100s | 100s |
| MountainCarContinuous-v0 | - | 500s |
| HalfCheetah-v3 | - | 2000s |

Given 200 NES outer loop iterations in the SE case and 50 steps in the RN case, these timings result in the following overall runtime for finding one SE or RN, respectively:

**SEs**:

- Acrobot: $\sim$ 6-7 hours
- CartPole: $\sim$ 5-6 hours

**RNs**:

- Cliff Walking: $\sim$ 8 minutes
- CartPole: $\sim$ 7 hours
- MountainCarContinuous: $\sim$ 7 hours
- HalfCheetah: $\sim$ 28 hours

We note that the required time can vary by more than a magnitude, depending on how quickly the RL agent can solve the environment in the inner loop.

## APPENDIX D   SUPERVISED LEARNING AND SYNTHETIC ENVIRONMENTS

One might ask how our proposed way of learning synthetic environments relates and compares to purely learning a synthetic environment model through a supervised learning (or model-based RL (MBRL)) objective. In this Section, we discuss the differences from our view point and show results of a small experiment that aimed at giving some answers to this question.

### D.1   CONTRASTING BOTH APPROACHES

First, our interpretation can be summarized with the following table:

We now elaborate in more detail what we believe are the core differences between both concepts and their resulting effects.

In the case of the supervised learning of the model as a jointly learned mapping from (state, action) pairs to the next state and the reward, we are limited by the human prior in regards to what is beneficial and informative to achieve a certain given task. This prior typically is encoded in the reward. Whereas the states of a problem almost always arise from a physical (e.g. Cartpole) or a logical/rule based (e.g. board games) foundation, the reward is to a much higher degree based on a human perspective. This limits the supervised approach to almost only a replication of the problem designer's choices. Now, it might be argued that therefore it is a pure approach for modeling a world or an environment, but by shifting the perspective towards the artificial nature of a reward signal,

Table 21: Comparing attributes between supervised learning and SE learning.

| | Synthetic Environments | Supervised Learning |
|---|---|---|
| **Learning Objective** | Mapping $(S,A) \rightarrow (S, R_{defined})$ | Mapping $(S, A) \rightarrow (S, R_{beneficial})$ |
| **Signal of Information** | Error in mapping like MSE, MLE | Performance of agent on given real environment |
| **Learned Function** | mapping defined by environment; quality in terms of closeness | A functional mapping of same input-output dimensionality as in the supervised case but defined to increase agents performance; Quality is in terms of reward on a given real environment |
| **Potential** | Imitate/replicate the given environment | Can imitate state transitions but is not restricted to this and can learn a very different environment that nevertheless helps to quickly train agents to do well on the real environment |

the choice of reward might as well be a source of misguidance or at least a potential hindrance to learning in a effective / efficient, and more unbiased fashion.

We believe that our approach reduces the potential shortcomings of human bias and choice of design to some degree by taking a more "task-aware" perspective. In our approach, the feedback signal for the model training comes from the given environment (similar to supervised model learning), but does not constrict the model to explicitly learn an approximate duplicate mapping. Rather, by using the performance of the agent as the only information, the model gains the ability to construct a mapping of states actions to the next states and a reward signal of which the sole purpose is to improve the future performance of the agent (task-awareness). With this, we allow the model to converge to a different kind of distribution (mapping) and potentially one that might squeeze or extend either or both state and rewards. This could be seen as a compression of information on the one hand and (by having a larger space) also as storing and extending additional information explicitly. Of course, our approach is not completely unbiased by the choice of reward because here as well the human-designed reward influences the performance signal on which the model is trained.

## D.2 COMPARING SUPERVISED LEARNING WITH SYNTHETIC ENVIRONMENTS ON CARTPOLE

We have conducted an experiment on the CartPole environment to compare SEs to models trained purely with supervised learning. Before we show results, we will explain our setup:

**Setup:** To train such a supervised learning model and also to maintain comparability to SEs, we used the data coming from SE runs (i.e. executing Alg. 1). More precisely, we recorded all the real environment data from each outer loop step on which agents are evaluated (10 test episodes), across 5 populations with each 16 workers/agents over 25-50 outer loop steps. The $(s, a, s', r)$ tuples coming from each population run were used to fit a supervised model, yielding in total 5 models. We note that the amount of data that each supervised model has been trained with corresponds to the same amount that we would use to learn an SE. The models have the identical architecture as the SEs and we used an MSE loss to optimize the error between predicted $(s', r)$ and ground truth $(s', r)$ pairs given $(s, a)$ as input. We trained the models for 100 epochs with Adam and a batch size of 1024, finally yielding an average error of 5e-07.

After fitting the supervised baseline models, we executed the same train and evaluation pipeline for SEs that we describe in Section 5. First, we trained agents with randomly sampled hyperparameters on the supervised model. Second, we evaluated them on the real environment and used the resulting rewards to fit the density plot curves. We hereby ensured the same number of reward data points (4000) as for the curves given in the paper by sampling more agent configurations. To also show transfer performance of the supervised baseline models, we executed the same procedure for unseen agents (Dueling DDQN & discrete TD3).

**Result:** In Figure 22 we can observe that the supervised baseline underperforms the other models significantly in terms of the reward distribution and the average cumulative reward (15.01 vs. 190.7 for SEs). In terms of training efficiency we can see a low number of steps and episodes but this

stems from inferior policies that cause an early termination. We also ran an experiment where we only considered two of the "best" models which we depict in Figure 23. Hereby "best" means that the majority of samples in the dataset come from high-quality trajectories achieving rewards $\geq 190$ (i.e. which effectively solve the environment). However, in this case, we can only observe a slight increase in performance compared to using all five models (15.01 vs 17.73 average cumulative reward).

The reasons for underperformance can be manifold and their analysis were not part of this experiment. Potential explanations for this are (but are not limited to):

- The supervised baseline models may be subject to overfitting, resulting in inaccurate dynamics models and rendering policy learning a difficult task (it also may increase agent hyperparameter sensitivity).

- Catastrophic forgetting / sensitivity to the order of seen trajectories during baseline training may not allow to recover accurate dynamics at, for example, early stages of policy learning.

- The supervised baseline models may entail other sources of inaccuracies, for example, smoothness or noise, resulting in an inferior approximation of the real environment that may also hinder policy learning.

- In the case of SEs, the task-aware meta-loss (and the meta-learning itself) may be a good regularizer for preventing the above failure cases.

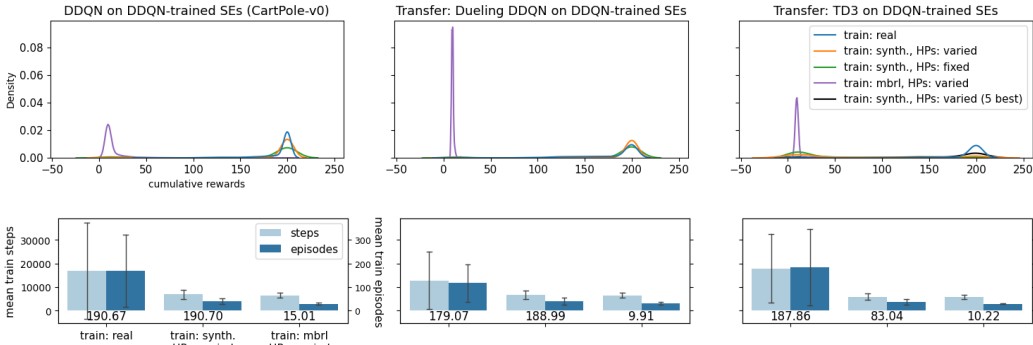

Figure 22: **Top:** Comparing the cumulative test reward densities of agents trained on SEs (green and orange), supervised baseline (purple), and baseline on real environment (blue). Agents trained on the supervised model underperform the SE models and the real baseline. **Bottom:** Comparing the needed number of test steps and test episodes to achieve above cumulative rewards. Left: real baseline, center: SEs, right: supervised (MBRL) baseline.

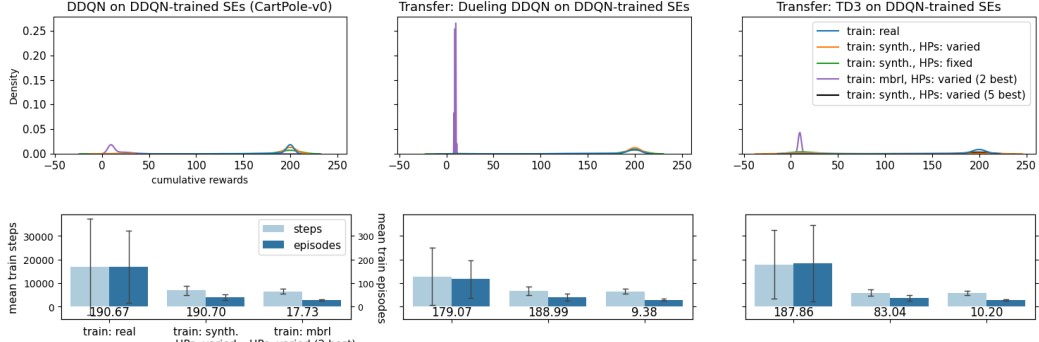

Figure 23: **Top:** Comparing the cumulative test reward densities of agents trained on SEs (green and orange), supervised baseline (purple), and baseline on real environment (blue). Agents trained on the supervised model underperform the SE models and the real baseline and using only the best two supervised models (purple) does not seem to change this. **Bottom:** Comparing the needed number of test steps and test episodes to achieve above cumulative rewards only using the best two of the five models. Left: real baseline, center: SEs, right: supervised (MBRL) baseline

## APPENDIX E  NES SCORE TRANSFORMATION

### E.1  SCORE TRANSFORMATION VARIANTS

The following overview lists eight score transformations for calculating $F_i$ based on the expected cumulative reward (ECR) $K_i$ or the associated rank $R_i$. These score transformations where also used in the hyperparameter optimization configuration space for finding SEs and RNs (e.g, in Tables 3 and 7). The rank $R_i$ is defined as the number of population members that have a lower ECR than the i-th member, i.e. the rank associated with the lowest ECR is 0, the one with the highest ECR $n_p - 1$ (the population size minus 1). We denote the ECR associated with the current SE or RN incumbent as $K_\psi$.

- **Linear Transformation**: The linear transformation maps all fitness values to the $[0; 1]$ range as

$$F_i = \frac{K_i - \min_i(K_i)}{\max_i(K_i) - \min_i(K_i)}, \tag{2}$$

  with $\min_i(K_i)$ as the lowest of all ECRs and $\max_i(K_i)$ as the highest of all ECRs.

- **Rank Transformation**: For the rank transform the individual rewards are first ranked with respect to each other, resulting in $R_i$. The ranks are then linearly mapped to the $[0; 1]$ range as

$$F_i = \frac{R_i - \min_i(R_i)}{\max_i(R_i) - \min_i(R_i)}, \tag{3}$$

  with $\min_i(R_i)$ as the lowest rank and $\max_i(R_i)$ as the highest rank.

- **NES**: Originally published in (Wierstra et al., 2008) and hence named after the paper "Natural Evolution Strategies" (NES), this method first transforms the rewards to their ranks $R_i$ in a similar manner as the rank transformation. In the next step, the fitness values are calculated from the individual ranks as

$$\hat{F}_i = \frac{\max\left(0, \log(\frac{n_p}{2} + 1) - \log(R_i)\right)}{\sum_{j=1}^{n_p} \max\left(0, \log(\frac{n_p}{2} + 1) - \log(R_j)\right)} - \frac{1}{n_p}, \tag{4}$$

$$F_i = \frac{\hat{F}_i}{\max_i\left(\left(|\hat{F}_i|\right)\right)}, \tag{5}$$

with $n_p$ as the size of the population. Whereas Eq. (4) constitutes the essential transformation, Eq. (5) acts solely as a scaling to the range $[x; 1]_{x<0}$ and is used for convenience. The NES transformation also allows negative fitness values $F_i$ through the $-\frac{1}{n_p}$ term in Eq. (4).

- **NES unnormalized**: We also consider an unnormalized NES score transformation variant. Eq. (4) and Eq. (5) here become

$$\hat{F}_i = \frac{\max\left(0, \log(\frac{n_p}{2}+1) - \log(R_i)\right)}{\sum\limits_{j=1}^{n_p} \max\left(0, \log(\frac{n_p}{2}+1) - \log(R_j)\right)}, \tag{6}$$

$$F_i = \frac{\hat{F}_i}{\max\limits_i\left(\left(|\hat{F}_i|\right)\right)}. \tag{7}$$

Although not directly visible from the equations above, the worse half of all fitness values becomes $0$.

- **Single Best**: We can induce a strong bias towards the best performing population member by only assigning a nonzero weight to the incumbent of the current ES iteration. The corresponding transformation scheme can be expressed by

$$F_i = \begin{cases} 1 \text{ if } R_i = \max\limits_i(R_i), \\ 0 \text{ else,} \end{cases} \tag{8}$$

with $\max\limits_i(R_i)$ as the best rank.

- **Single Better**: It might be advantageous to update the SE and RN parameters only if their performance is supposed to improve. By calculating the ECR $K_\psi$ associated with the parameters, the adapted "single best" transformation becomes

$$F_i = \begin{cases} 1 \text{ if } K_i > K_\psi \text{ and } K_i = \max\limits_i(K_i), \\ 0 \text{ else.} \end{cases} \tag{9}$$

- **All Better 1**: The "single best" transformation relies at most on a single population member to update the SE or RN parameters in each ES iteration. A more regularized version considers all population members whose ECR $K_i$ is better than the ECR $K_\psi$ of the current SE or RN parameters. In a more mathematical notation, this can be expressed as

$$\hat{F}_i = \begin{cases} \frac{K_i - K_\psi}{\max\limits_i(K_i) - K_\psi} \text{ if } K_i > K_\psi, \\ 0 \text{ else,} \end{cases} \tag{10}$$

$$F_i = \frac{\hat{F}_i}{\max\limits_i \hat{F}_i}, \tag{11}$$

with $K_\psi$ being evaluated similarly as the individual ECRs $K_i$.

- **All Better 2**: This transformation is almost identical to the "all better 1" transformation, except that we divide by the sum of all $\hat{F}_i$ values when calculating the $F_i$ values and not by the maximum $\hat{F}_i$ value:

$$\hat{F}_i = \begin{cases} \frac{K_i - K_\psi}{\max\limits_i(K_i) - K_\psi} \text{ if } K_i > K_\psi, \\ 0 \text{ else,} \end{cases} \tag{12}$$

$$F_i = \frac{\hat{F}_i}{\sum\limits_i \hat{F}_i}, \tag{13}$$

## E.2 EVALUATION OF SCORE TRANSFORMATION VARIANTS

Below we visualize the BOHB hyperparameter optimization results of different score transformation variants for SEs and plot these as a function of the NES step size with (left) and without (right)

mirrored sampling. The colors indicate the number of outer loop iterations of Algorithm 1 required to solve the target task (here: 2x2 grid world environment). Green signifies 0, yellow 25, and red 50 outer loop iterations. Each of the roughly 3500 dots corresponds to a single evaluation. We can observe multiple things:

- none of the rank transformations fails completely for the given simple environment

- mirrored sampling is only marginally better (if at all) while doubling the computational load (at least in this simple 2x2 grid world environment)

- noise: good performing samples in regions with bad performing samples and vice versa

- the "all better " score transformation seems to perform best

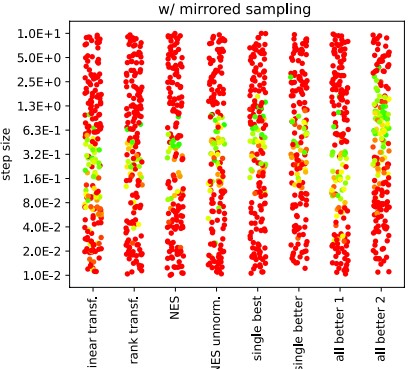 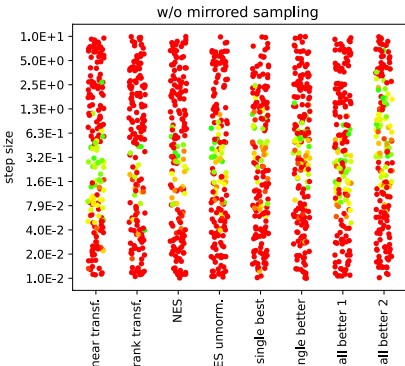

Figure 24: Evaluation of different score transformation schemes for synthetic environments: Shown is the required number of NES outer loop iterations (green: 0, yellow: 25, red: 50) for different hyperparameter configurations until an SE is found that solves the real environment (here: a 2x2 grid world environment).

# APPENDIX F  ADDITIONAL PLOTS

## F.1  SYNTHETIC ENVIRONMENTS

### F.1.1  EVALUATION OF INDIVIDUAL SE MODELS (CARTPOLE)

Below we show the individual or model-based violin plots for the 40 SE models which we calculated based on the cumulative rewards of 1000 randomly sampled agents that were each evaluated across 10 test episodes. For agent sampling we again chose the hyperparameters ranges denoted in Table 2. As the best five models for discrete TD3 (Figure 3) we chose the models `CartPole-v0_25_4I271D`, `CartPole-v0_22_CW9CSH`, `CartPole-v0_21_DRWNIK`, `CartPole-v0_8_QGN9IF`, and `CartPole-v0_17_WPWK3R`.

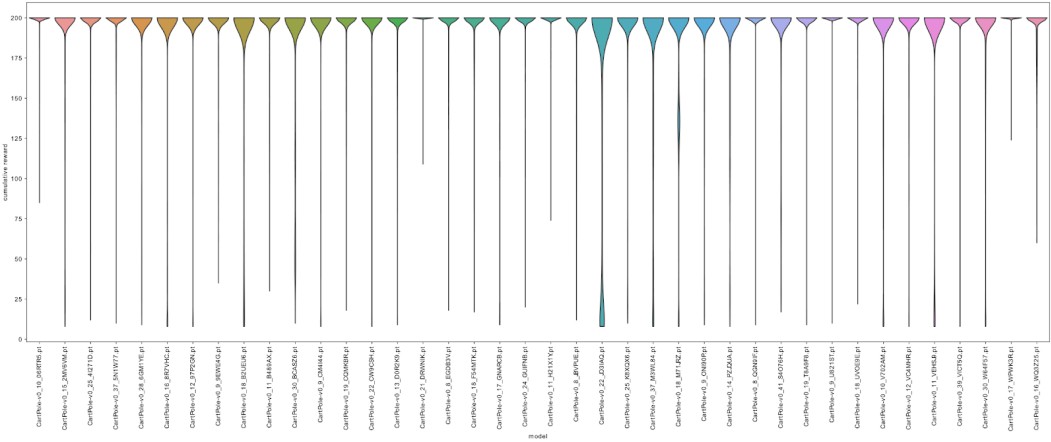

Figure 25: Model-wise evaluation: 1000 randomly sampled **DDQN** agents à 10 test episodes per SE model.

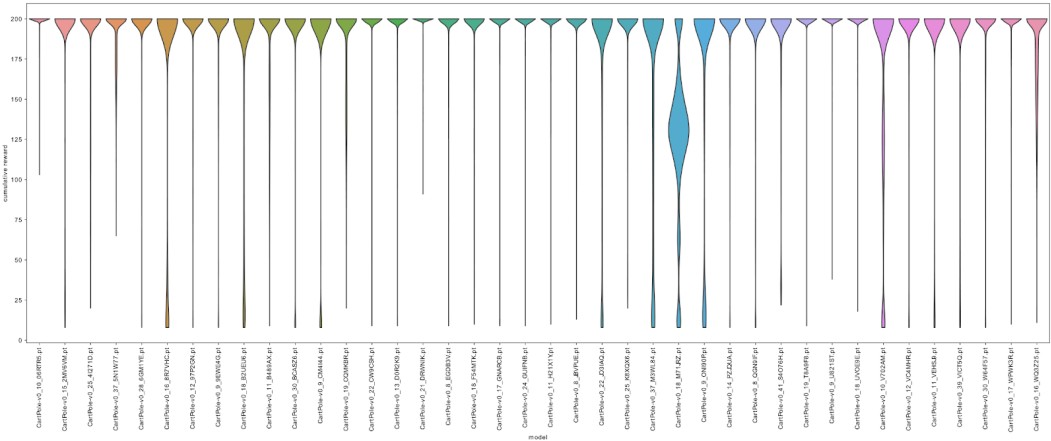

Figure 26: Model-wise evaluation based on the cumulative rewards of 1000 randomly sampled **Dueling DDQN** agents à 10 test episodes per SE model.

#### F.1.2 EVALUATION OF INDIVIDUAL SE MODELS (ACROBOT)

Below we show the individual or model-based violin plots for the 40 SE models which we calculated based on the cumulative rewards of 1000 randomly sampled agents that were each evaluated across 10 test episodes. For agent sampling we again chose the hyperparameters ranges denoted in Table 2. As the best five models for discrete TD3 (Figure 7) we chose the models `Acrobot-v1_223R5W`, `Acrobot-v1_2XMO40`, `Acrobot-v1_7DQP5Z`, `Acrobot-v1_9MA9G8` and `Acrobot-v1_8MYN67`.

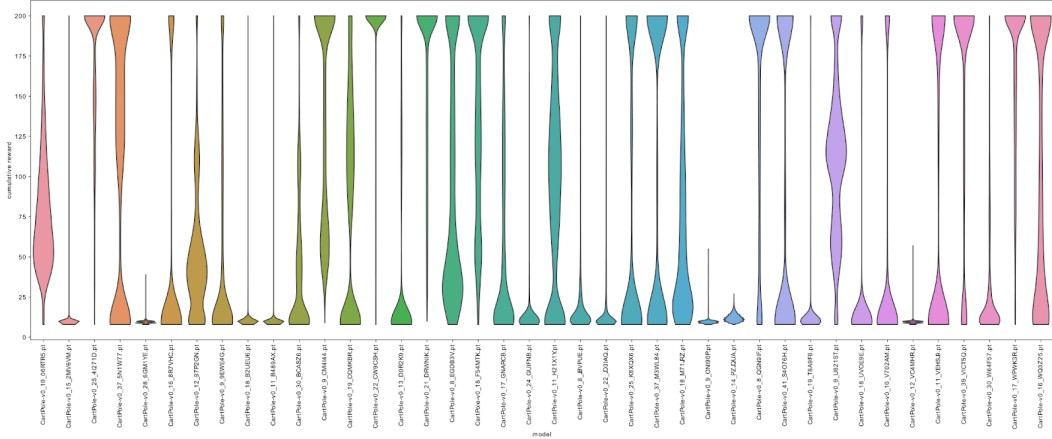

Figure 27: Model-wise evaluation based on the cumulative rewards of 1000 randomly sampled **discrete TD3 agents** à 10 test episodes per SE model.

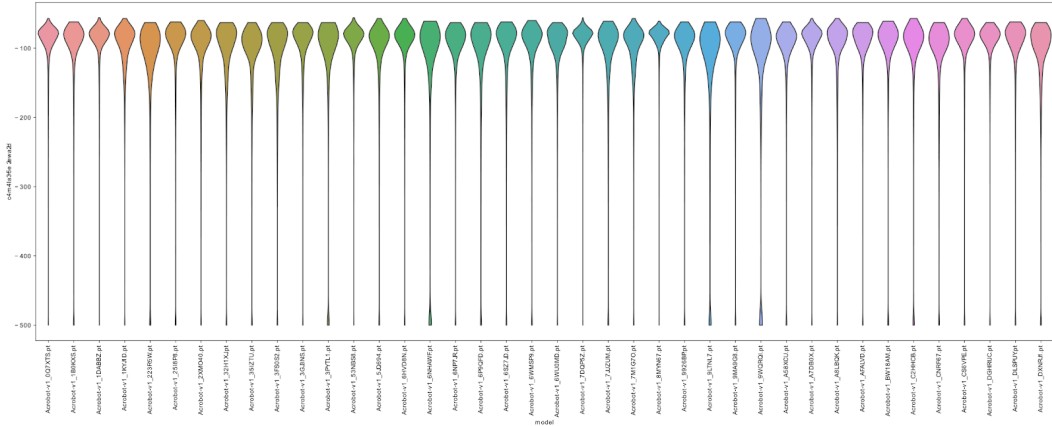

Figure 28: Model-wise evaluation based on the cumulative rewards of 1000 randomly sampled **DDQN** agents à 10 test episodes per SE model.

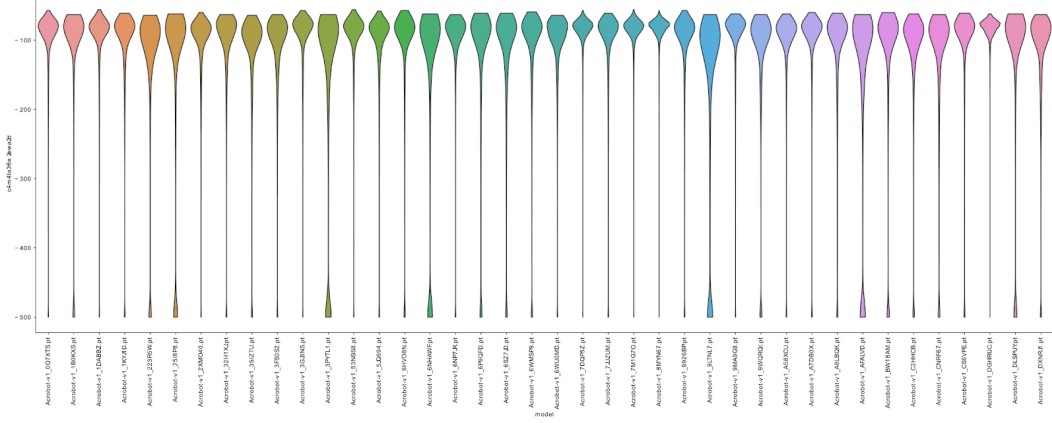

Figure 29: Model-wise evaluation based on the cumulative rewards of 1000 randomly sampled **Dueling DDQN** agents à 10 test episodes per SE model.

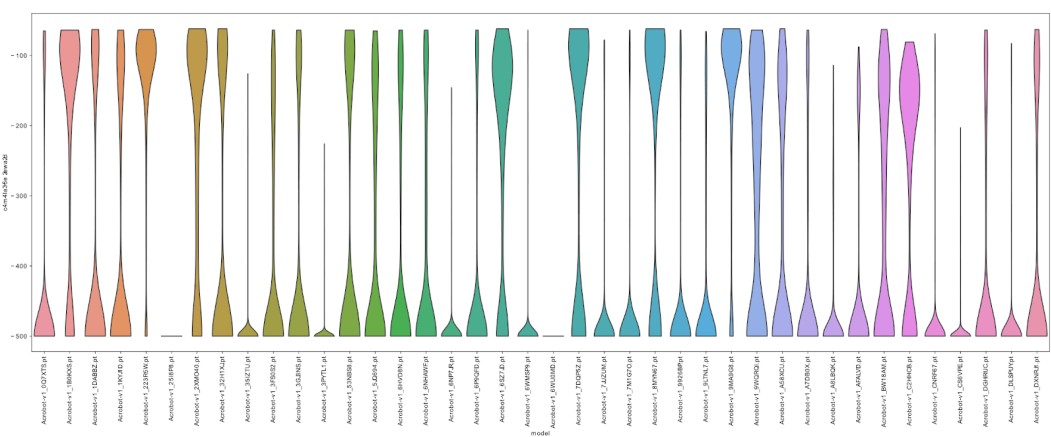

Figure 30: Model-wise evaluation based on the cumulative rewards of 1000 randomly sampled **discrete TD3 agents** à 10 test episodes per SE model.

