# OpenReview forum: "Learning Synthetic Environments and Reward Networks for Reinforcement Learning"
_ICLR.cc/2022/Conference — ICLR 2022 Poster_

### Official Review · Reviewer_ktr8 · 2021-10-30

**Correctness:** 3
**Technical Novelty And Significance:** 3
**Empirical Novelty And Significance:** 3
**Recommendation:** 6
**Confidence:** 4

**Main Review:**

Results:
- In Figure 2, the performance seems very unstable: In particular, the reward for individual agents seems to fluctuate widely and even the average starts drooping after the 50th iteration in the arcobot task. It would be great to know what the source of this instability is.

Conceptual:
- There are repeated claims in the paper that the synthetic environment "focuses the learning" on the relevant states or learns "representations" of the target environment ("an informed representation of the target environment by
modifying the state distributions to bias agents towards relevant states"). This claim is entirely unclear to me at the moment. In particular, there simply doesn't seem to be any reason that the SE should have much in common with the ground truth environment. The only condition is that _optimal policies_ for the two environments match. In particular, there will be many environments in which the optimal policy is likely much easier to learn than in the original Env.
For example, you can imagine a state transition dynamic that is entirely random but provides a large reward for the optimal action in each of the given states. Another option is to learn environment dynamics where the optimal action in each state deterministically transitions to an arbitrary next state, such that the agent will see all required states during training.
It would be great to provide any type of evidence that there is actually semantic meaning in the SE that matches the underlying ground truth environment.

Other comments:
- In Figure 2, it seems like the legend is switched? I.e. dashed line should be the solved threshold?

**Summary Of The Paper:**

This paper introduces Synthetic Environments and Synthetic Rewards as a bi-level optimization problem. The main idea is to find SEs and REs such that an RL agent trained under them will do well in a given, known target environment, without further training.

**Summary Of The Review:**

A very interesting paper with early results in a potentially interesting direction. It is currently unclear what this method allows to do that was previously impossible from a practical point of view, but there seems to be promise.
I'd be open to raising my score if my concerns are addressed by the authors.



Update:
Based on the discussion phase and the rebuttal I have updated the score to a 6. I believe this paper has enough merit to be published at ICLR, if there is space. It is an intriguing piece of work, even though the practical utility is very unclear at this point.

---

> ### Author Response · Authors · 2021-11-13
> **Response to Reviewer ktr8**
>
> > In Figure 2, the performance seems very unstable: In particular, the reward for individual agents seems to fluctuate widely and even the average starts drooping after the 50th iteration in the arcobot task. It would be great to know what the source of this instability is.
>
> Thank you very much for pointing out this concern. We are sorry for not having discussed the reason for this in the main paper. Due to the space limitations we only addressed this in the appendix (Appendix A.2), where we point out that the variance arises from the stochasticity of natural gradients and from the sensitivity of RL agents to seeds and parameter initializations. If you think we should move this information to the main manuscript we would be very happy to do so.
>
> \
> &nbsp;
>
> > There are repeated claims in the paper that the synthetic environment "focuses the learning" on the relevant states or learns "representations" of the target environment ("an informed representation of the target environment by modifying the state distributions to bias agents towards relevant states"). This claim is entirely unclear to me at the moment. In particular, there simply doesn't seem to be any reason that the SE should have much in common with the ground truth environment. The only condition is that optimal policies for the two environments match. In particular, there will be many environments in which the optimal policy is likely much easier to learn than in the original Env. For example, you can imagine a state transition dynamic that is entirely random but provides a large reward for the optimal action in each of the given states. Another option is to learn environment dynamics where the optimal action in each state deterministically transitions to an arbitrary next state, such that the agent will see all required states during training. It would be great to provide any type of evidence that there is actually semantic meaning in the SE that matches the underlying ground truth environment.
>
> We are sorry to read that our made claims remain unclear to you. We thank you for sharing these concerns with us which we would like to address now. Most importantly, we want to emphasize that our optimization scheme does not bias the SEs to entail the true dynamics or reward or even have semantic meaning in accordance with the physical laws prevalent in the ground truth environment. It is because of this, that we believe that in our experiment results we do indeed not see evidence for semantic meaning that matches the ground truth environment.
>
> Here, we would also kindly refer to Appendix D where we study the case when training SEs with the objective of imitating the ground truth environment in a supervised learning (or model-based RL) setup. We believe the discussion given there may exactly address your doubts and clarify your concerns. Please let us know if anything else is unclear and requires further details or discussion. We would also be very happy to move missing information from the appendix to the main manuscript if you believe it would increase general understanding of what SEs learn.
>
> \
> &nbsp;
>
> > In Figure 2, it seems like the legend is switched? I.e. dashed line should be the solved threshold?
>
> Thank you for this remark. Indeed, the legend entries were switched; we have corrected this and updated the paper.
>
> If we addressed your concerns, would you be kind enough to raise your score?

---

### Official Review · Reviewer_Q4vZ · 2021-11-02

**Correctness:** 3
**Technical Novelty And Significance:** 2
**Empirical Novelty And Significance:** 2
**Recommendation:** 3
**Confidence:** 3

**Main Review:**

### Strengths
- This paper extends prior work on meta-learning MDP transition dynamics for improving training performance on a target environment, by introducing a method that aims to learn both the state and reward transitions. This formulation is more general than prior works.
- Experimental results are provided for multiple formulations of meta-learned MDP transition dynamics, spanning both joint state and reward networks (SEs) as well as various formulations of a reward network (potential and non-potential-based reward shaping parameterizations).
- This paper includes experiments on transferring to different agent architectures and hyperparameter settings, which seems to be a novel experimental setting for meta-learning MDP dynamics.

### Weaknesses
- The motivation for learning SEs and RNs for singleton target environments is not convincing. While there seem to be some marginal gains in sample-efficiency, it would seem these gains are largely made irrelevant due to the additional training needed to train the environment in the first place. Moreover, if SEs and RNs require first training the agent multiple times on synthetic environments, why not just copy weights or use a form of policy distillation, e.g. kickstarting (Schmitt et al, 2018), rather than train the agent again? Further, it seems the basic ICM baseline already matches or outperforms the proposed RN approaches across experimental settings, further weakening the argument for using the more computationally expensive method proposed in this work.
- Given the wide design space of possible ways of learning the state and reward transition dynamics for a target real environment, it is unclear why only these two (SE and RN) formulations were studied. Given that this is an empirical paper, the value of the findings rests largely on a clean experimental design that provides a convincing recommendation based on comprehensive experiments. The seemingly arbitrary choice for only considering the SE and RN formulation does not provide such a convincing recommendation for using either of these formulations.
- RNs are effectively an ablation of SEs, where the state is not learned. However, the current experimental design does not neatly present these results as a clean ablation. This is because the RN experiments also separately consider various kinds of potential and non-potential-based reward shaping parameterizations for the learned reward. These parameterizations are not studied in combination with SEs.
- Likewise, there is no corresponding ablation for when the reward is not learned, but only the state is learned, i.e. a "state-transition network" or perhaps a "state-augmentation network."
- It is not made clear why exactly SEs and RNs have to use different objectives when training.
- The RN objective seems strange, as it seems to be optimizing for sample efficiency rather than performance. In contrast the SE objective optimizes for performance. This seems like a significant inconsistency between the training methology for the two types of models, making it hard to compare the relative merits of SEs and RNs.
- The reward difference term of the RN objective also caps the maximum performance improvement compared to training on the real environment. The motivation for this objective design is not clear, as not taking the max between the reward difference and 0 should encourage learning RNs that lead to outperforming training on the real environment, rather than simply matching it.
- Further, it seems that there is not a consistent ranking in terms of which parameterization for the RNs improves training performance on the real environment the most, and similarly, no consistent ranking between non-transfer and transfer scenarios for each environment.
- Meanwhile, the SE results on CartPole and Acrobot would benefit from a direct comparison to the training curves for agents using the best hyperparameters, trained on the real environment. This would directly separate out the performance gains due to training on the  SE.
- The comparison to previous work such as Zheng et al, 2020 seems quite hand-wavey. Zheng et al, 2020 in particular is quite similar to this work in terms of analyzing the effect of learning an RN and also includes more comprehensive experimental results in a more complex setting requiring recurrent RNs. The novel contributions in relation to these prior works seems to be around analyzing the impact of varying agent hyperparameters on transferring to alternative agent settings and architectures during evaluation. However, these transfer results do not seem consistently strong for any one method proposed.
- Sampling agent hyperparameter settings from a diverse range seems crucial for the learned SEs and RNs to transfer to other agent configurations. The paper does not discuss how sensitive the proposed methods are to the agent hyperparameter ranges used during training.

**Summary Of The Paper:**

This paper aims to learn proxy environments (synthetic environments or SEs) and reward functions (reward networks or RNs), parameterized as neural networks, such that these proxy models provide beneficial transitions to make it more sample-efficient to learn a policy for a fixed target environment (referred to as the real environment). SEs replace the observed state and rewards during training, while reward networks provide a synthetic reward that is a function of the real reward, current state, and next state. The proposed method formulates this problem as a bi-level optimization, where the inner loop consists of standard RL under the proxy model, and the outer loop consists of NES with the aim of optimizing either the performance on the true, target environment (SEs) or a the number of training steps needed to reach a certain return threshold on the real environment.

**Summary Of The Review:**

While this work proposes an interesting generalization of prior work on meta-learning MDP dynamics, the case made for these methods is weak, and the experimental results—especially for SEs—are uncompelling. Given the additional compute required by these methods, it seems that various forms of policy distillation or even training from scratch seem more efficient. Further, it is not clear why only certain formulations (SEs and RNs) of meta-learning neural transition dynamics were investigated, while others (like a state-only neural transition network) were not. Given these points, I recommend this paper in its current form for rejection.

---

> ### Author Response · Authors · 2021-11-13
> **Response to Reviewer Q4vZ (Part 1)**
>
> > This paper extends prior work on meta-learning MDP transition dynamics for improving training performance on a target environment, by introducing a method that aims to learn both the state and reward transitions. This formulation is more general than prior works.
>
> We are thankful for your comment. To better understand your perspective with regards to the below mentioned weaknesses, we would kindly ask you for references on prior work on meta-learning MDP transition dynamics that you would say our work extends. From our point of view, we are the first that explore meta-learning neural networks as proxies for transition dynamics of a target environment. We elaborated on this point extensively in the related work section and we would therefore greatly appreciate any pointers to prior work that we might have missed.
>
> \
> &nbsp;
>
> > This paper includes experiments on transferring to different agent architectures and hyperparameter settings, which seems to be a novel experimental setting for meta-learning MDP dynamics.
>
> Thank you for acknowledging our experimental contribution. To avoid confusion, we would like to kindly emphasize that, instead of studying the transfer to agent architectures, we studied the transfer to multiple agent algorithms.
>
> \
> &nbsp;
>
> > The motivation for learning SEs and RNs for singleton target environments is not convincing. While there seem to be some marginal gains in sample-efficiency, it would seem these gains are largely made irrelevant due to the additional training needed to train the environment in the first place. Moreover, if SEs and RNs require first training the agent multiple times on synthetic environments, why not just copy weights or use a form of policy distillation, e.g. kickstarting (Schmitt et al, 2018), rather than train the agent again?
>
> Thank you for communicating your concerns with us. While we agree that learning SEs is expensive, we respectfully disagree that improvements of 30-65% can be deemed marginal. More important to point out is that these gains are achieved without explicitly optimizing for it, which, in our opinion, is a very interesting finding. Moreover, we believe that focussing mostly on the inefficiency of our method does our work a disservice since it disregards the many other gains, for example, its ability to transfer to other agents and to train these efficiently or the possibility to do task analysis (Analyzing SE Behavior, Section 5.1) and interpretation. We did not intend to make a claim of end-to-end speedups; rather, we find it fascinating that it is possible to learn synthetic environments *at all* that allow for faster training, even of other agents. That finding is indeed one of the key contributions of our paper, not the particular (granted, quite costly) evolutionary algorithm for doing so.
>
> Lastly, while we agree that loading agent weights from earlier outer loop iterations *could* make the optimization more efficient, this falls into the category of a speedup technique for the fundamentally novel task we’re proposing in this paper. While this may indeed lead to speedups, we believe that in the first paper on SEs we should rather scientifically study SEs (like in our analysis) without directly introducing confounding factors due to a speedup technique. In particular, it is unclear to us a priori how the proposed speedup technique would affect the teaching capacity of SE models. For example, the efficiency gains could be lost since the SE may not learn to properly teach agents from scratch. As mentioned above in our response to Reviewer 1Zys, we do, however, believe that loading pre-trained weights is a good idea and we are considering it for future work on learning SEs.
>
> All in all, while we share many of your concerns, we would appreciate it if we could work jointly together to improve the motivation and communication of our conclusions to make them clearer for you.
>
> \
> &nbsp;
>
> > Further, it seems the basic ICM baseline already matches or outperforms the proposed RN approaches across experimental settings, further weakening the argument for using the more computationally expensive method proposed in this work.
>
> We respectfully, but firmly, disagree here. The ICM baseline does not outperform the RNs. More specifically, in terms of 1) efficiency, the ICM never outperforms any RN variants and in terms of 2) final performance, the ICM only outperforms the RNs for one case (the Cliff environment in the varied hyperparameter setting, see Fig. 5).

---

> ### Author Response · Authors · 2021-11-13
> **Response to Reviewer Q4vZ (Part 2)**
>
> > Given the wide design space of possible ways of learning the state and reward transition dynamics for a target real environment, it is unclear why only these two (SE and RN) formulations were studied. Given that this is an empirical paper, the value of the findings rests largely on a clean experimental design that provides a convincing recommendation based on comprehensive experiments. The seemingly arbitrary choice for only considering the SE and RN formulation does not provide such a convincing recommendation for using either of these formulations.
>
> We welcome your comments on a potential downside in our experiment design. Overall, we think there are three choices of formulations to be considered. One can model the reward dynamics, the state dynamics or both. In our paper, we chose the former and latter which, in our point of view, does not constitute an arbitrary, but a logical choice. We are a bit puzzled by the statement about our experiment design and would appreciate it if you could make this more concrete. We would be very happy to engage in a discussion about any formulation that you would find more interesting to study and that would lead to a better experiment design.
>
> \
> &nbsp;
>
> > RNs are effectively an ablation of SEs, where the state is not learned. However, the current experimental design does not neatly present these results as a clean ablation. This is because the RN experiments also separately consider various kinds of potential and non-potential-based reward shaping parameterizations for the learned reward. These parameterizations are not studied in combination with SEs.
>
> We find your view point of RNs being an ablation of SEs very interesting and thank you for this input. While it is an interesting idea to present RNs as an ablation of SEs, we are unsure why phrasing RNs as a subset of the broader formulation of SEs would constitute a weakness. We would be happy to engage in a discussion with you about this.
>
> In terms of the different parameterizations not being studied in the SE case, we believe there may be a misunderstanding. In particular, we want to point out that in the SE formulation we do not want to depend on rewards from the real environment but rather have them artificially generated. Consequently, by adopting the RN parameterization variants from Table 1 to the case of SEs, one would effectively end up only with the two variants denoted as “Exclusive” to test for the SEs which do not depend on the real reward. However, the variant denoted as “Exclusive Non-potential RN” from these two variants is essentially the SE case (with the addition that we ask the SE to produce artificial states along the artificial rewards). So this can actually be seen as an ablation of SEs without the dynamics component. The other variant we are left with is “Exclusive Potential RN”. We acknowledge that testing this parameterization is indeed interesting in the SE case but we also point out that it does not fully follow the potential-based reward shaping theory as it does not augment a reward but produces a fully artificial one. This is the reason why we did not pursue this direction of research when studying SEs. Another reason why we did not investigate PBRS for SEs is that the theory usually provides guarantees only for reward augmentations and not artificial states.
>
> We hope we were able to address your concerns and are open to engaging in a discussion on this.
>
> \
> &nbsp;
>
> > Likewise, there is no corresponding ablation for when the reward is not learned, but only the state is learned, i.e. a "state-transition network" or perhaps a "state-augmentation network."
>
> Thank you for this suggestion. We actually tried this, but since in this case the state transitions differ, the original reward signal does not capture any meaning, and the reward rather would need to be adapted to the different state transitions (as in full SEs).

---

> ### Author Response · Authors · 2021-11-13
> **Response to Reviewer Q4vZ (Part 3)**
>
> > It is not made clear why exactly SEs and RNs have to use different objectives when training.
>
> > The RN objective seems strange, as it seems to be optimizing for sample efficiency rather than performance. In contrast the SE objective optimizes for performance. This seems like a significant inconsistency between the training methology for the two types of models, making it hard to compare the relative merits of SEs and RNs.
>
> > The reward difference term of the RN objective also caps the maximum performance improvement compared to training on the real environment. The motivation for this objective design is not clear, as not taking the max between the reward difference and 0 should encourage learning RNs that lead to outperforming training on the real environment, rather than simply matching it.
>
>
> Thank you for opening the discussion about the RN objective. We agree with you that this detail may be confusing to readers. Frankly speaking, the choice of this objective is not greatly motivated. We implemented this objective after we noticed the efficiency improvements from SEs. To test whether RNs could also yield such efficiency improvements, we chose to directly optimize for efficiency. Another reason for optimizing for efficiency directly was that the real environment is involved in the RN case and we pursued the goal to reduce real environment interactions.
>
> Nevertheless, we now conducted a small study to contrast different objectives in the RN case and made the results available here:
>
> https://docs.google.com/document/d/1LLshom6G8gr-Si6-RKPYfnhHM7BYrCeKhkwEJn2BTrk/edit?usp=sharing
>
> The overall conclusion from these results is that an AUC objective generally yields similar results as for the reward threshold objective, but in some cases it seems to reduce efficiency a bit. However, the overall significant efficiency gains remain. Moreover, in contrast to SEs, efficiency improvements in the RN case seem to arise more often when explicitly optimized for it (see Figures under “AUC” or “reward threshold”) while maximizing reward (only tested for CartPole and Cliff environments) leads to efficiency improvements only in the CartPole but not in the Cliff environment
>
> We hope by providing these results, we are able to convince you that a relative comparison between SEs and RNs is still possible. Did we address your concerns appropriately? Again, we would be happy to engage in a discussion with you.
>
> \
> &nbsp;
>
> > Further, it seems that there is not a consistent ranking in terms of which parameterization for the RNs improves training performance on the real environment the most, and similarly, no consistent ranking between non-transfer and transfer scenarios for each environment.
>
> This observation is correct. We share and describe it in our paper. However, it is unclear to us how this is a weakness of our work. We believe this is certainly something to study in more detail in the future.
>
> \
> &nbsp;
>
> > Meanwhile, the SE results on CartPole and Acrobot would benefit from a direct comparison to the training curves for agents using the best hyperparameters, trained on the real environment. This would directly separate out the performance gains due to training on the SE.
>
> Thank you for your comment. We believe the choice of hyperparameters is irrelevant as long as the identical hyperparameter configuration is used on the SE and the real environment to enable a fair comparison. In our case, we chose the default ones from the original papers. In fact, we have executed the SE performance experiment with optimal agent hyperparameters in both cases, i.e. agents trained on SEs and on the baselines and the results were identical, meaning that the same performance gaps were observable.
>
> \
> &nbsp;
>
> > The comparison to previous work such as Zheng et al, 2020 seems quite hand-wavey. Zheng et al, 2020 in particular is quite similar to this work in terms of analyzing the effect of learning an RN and also includes more comprehensive experimental results in a more complex setting requiring recurrent RNs.
>
> We again respectfully, but firmly, disagree here. As described in our related work section, Zheng et al. explore learning reward networks for gridworld-like environments only. Thus, we believe our continuous-control environments constitute a much more complex setting. More importantly, however, is that Zheng et al. focus on learning a reward as a distribution over tasks and in contrast to our work not on a single task.
>
> Moreover, their method facilitates exploration across episodes by taking into account the entire lifetime of an agent with the goal to distill knowledge about long-term exploration and exploitation into a reward function. Therefore, they not only consider a different objective but also their goal is different to our paper. All in all, we think that their work studies entirely different research questions.

---

> ### Author Response · Authors · 2021-11-13
> **Response to Reviewer Q4vZ (Part 4)**
>
> > The novel contributions in relation to these prior works seems to be around analyzing the impact of varying agent hyperparameters on transferring to alternative agent settings and architectures during evaluation. However, these transfer results do not seem consistently strong for any one method proposed.
>
> As mentioned above, we are convinced that the novel contributions in relation to Zheng et al. revolve around: 1) learning state dynamics (SEs), 2) using more complex environments in the RN case, 3) a much richer analysis of agent transfer (SEs and RNs), and 4) showing robustness to hyperparameter variations (SEs and RNs). We agree with you that the transfer performance is only high within an agent family but we still see this particular result as intriguing. In particular, we want to mention that we did not apply any “tricks” nor did we put any effort into improving transfer performance but our results nevertheless indicate that transfer is possible.
>
> \
> &nbsp;
>
> > Sampling agent hyperparameter settings from a diverse range seems crucial for the learned SEs and RNs to transfer to other agent configurations. The paper does not discuss how sensitive the proposed methods are to the agent hyperparameter ranges used during training.
>
> Thank you for sharing this concern with us. We believe there may be an overestimation of the influence of varying the hyperparameters during the learning of SEs and RNs. First, one can see in Fig. 3 that the difference between varying hyperparameters and not varying hyperparameters during SE training is low (compare “train: synth. HPs: varied” with “train: synth. HPs: fixed”). Second, in the transfer case of teaching entirely new agents, we want to note that we evaluate the transferability of RN and SEs under very challenging conditions since we use completely unseen agents that also come from entirely different RL agent families (e.g. off-policy to on-policy or q-learning to actor-critic). Here, we strongly doubt that making the hyperparameter range more diverse during SE and RN training will adequately “prepare” an SE or RN model to transfer to an entirely different agent type.
>
> Moreover, we discuss the sensitivity of hyperparameter variation during SE/RN training repeatedly in our paper. Not only do we analyse the behavior of SE learning without varying the agent hyperparameters in Fig. 3, we also discuss this in several parts of the paper. Here are a few excerpts:
> * “Training DDQN agents on DDQN-trained SEs without varying the agent’s HPs (green) during SE training clearly yielded the worst results. We attribute this to overfitting of the SEs to the specific agent HPs. Instead, when varying the agent HPs (orange) during SE training” (Section 5.1; Evaluating the Performance of SE-trained Agents)
> * “As before, not varying the DDQN agent HPs during SE training reduces transfer performance (green).” (Section 5.1; Transferability of SEs)
> * “Contrary to training SEs, we do not vary the agent HPs during RN training as we did not observe improvements in doing so.” (end of Section 4)
> * “Like in the CartPole case, we show three different SE training settings: agents trained on real env. with varying agent HPs (blue), on DDQN-trained SEs when varying agent HPs during NES runs (orange), on DDQN-trained SEs where the agent HPs were fixed during training of the SEs (green).” (Appendix A.3)
>
> Lastly, the choice of the diversity of our hyperparameter configurations is easily explained: the chosen lower value range for the sampled hyperparameters (Table 2) is simply equivalent to the default lower bound divided by three and similarly for the higher value range (multiplied by three).
>
> Please let us know if the presentation of this information needs to be changed to be more accessible to a reader. We hope we were able to address your concerns appropriately and would be grateful to engage in a discussion with you about any remaining concerns. An indication about what it would take to convince you of our work’s merit would also be highly appreciated.

---

> > ### Comment · Reviewer_Q4vZ · 2021-11-24
> > **Thanks for the response**
> >
> > I appreciate the authors' response. However, my main concern remains around the motivation for this setting and the additional insights with respect to prior work. I believe the results around transfer to novel agents via domain randomization over hyperparameters seems most interesting in justifying investment in this approach, and would encourage the authors to double down on this aspect of their findings in a future submission of this work. The current state of results around this aspect of the work seems inconsistent.

---

> > > ### Author Response · Authors · 2021-11-30
> > > **In hope of clarification**
> > >
> > > Dear reviewer,
> > >
> > > Considering the breadth and depth of our initial response and your relatively short response in return, we would like to point out that only through *actionable* feedback we are able to address your concerns successfully and ultimately improve our paper. Given your response, we are unsure about what would be steps to mitigate your concerns.
> > >
> > > We are convinced that our proposed method is well-motivated and the evaluation is sound. We tried to address your concerns in detail above but your response did not touch on this at all. Therefore, we would kindly ask you to explain in detail whether, from your perspective, there exists a problem with the method or the types of evaluations we have studied.
> > >
> > > Thank you.

---

> > > > ### Comment · Reviewer_Q4vZ · 2021-11-30
> > > > **Please see previous comment**
> > > >
> > > > Please see my previous comment, which highlighted the portions of my original review that your response failed to address.

---

### Official Review · Reviewer_pG56 · 2021-11-03

**Correctness:** 4
**Technical Novelty And Significance:** 3
**Empirical Novelty And Significance:** 4
**Recommendation:** 8
**Confidence:** 3

**Main Review:**


First and foremost, let me start by saying that I *really* like this manuscript.

1. The writing is extremely clear & detailed, and makes the paper a pleasure to read. I also found that the authors tended to answer questions that were popping as I was reading paragraphs immediately thereafter, which is a likely sign of writing that has gone through a lot of careful iterations.

2. The authors have gone through a lot of work to justify the choice of optimization methods, how and why the framework was setup in certain ways, and how the hyperparameters for a now fairly complex learning system were chosen. Considering the code release and the extremely detailed appendices, I suspect the work will be extremely easy to reproduce. Truly commendable, and a great example of what good ML research looks like (I also particularly enjoyed the frank discussion about the work's limitation due to choosing ES as an optimization method for the outer loop).

3. The central idea of the method is relatively simple, but seems to be quite effective; it honestly borrows from a lot of previous literature (reward learning, population-based optimization, etc.), whilst effectively producing a novel take on learning syntethic environment models.

So, overall, I am extremely happy to argue for acceptance as it is. Now, let's look at what I perceive to be some of its weaknesses:

4. The manuscript at times feels very dense, especially when looking at methodology and experimental details, and its understanding is greatly helped from the presence of a sizable appendix. It very much feels like the authors could have a feasibly split the paper into two, one on ES and a follow up on RN, providing more space to experimentally analyse ES and RN separately and make for a cleaner and easier review process.

5. Some of the hypotheses presented to justify the improved agent training feel relatively handwave-y. Consider for instance the following quote (emphasis mine):

>Optimizing SEs for the maximum cumulative reward automatically yielded fast training of agents as a side-product, likely due to **efficient synthetic state dynamics**.

It is implied that this meta-learning approach leads to generally discover better data-producing dynamics for the agents used in the system, but I wonder whether this is primarily due to the structure of the tasks employed in the experimental settings (as state space and transition dynamics of these simple control problems are _generally_ not very informative, and the _useful_ -- policy learning wise -- parts of these MDP are instead both easily discoverable via search and extremely informative).

In short, I wonder if we'd see similar improvements for tasks / environments where task difficulty is extremely affected by the emerging complexity of the environment, or whether we can actually show that this is a property of SEs / RNs when trained in this manner (which would be quite outstanding!). Although that said, the effecitveness of RNs do seem to indeed point to the latter hypothesis.


### Nits / Additional comments

Figure 2: the labels are inverted (dotted lines should be the thresholds).

Section 6 (but generally in many parts of the paper): it is claimed that learning rewards is generally easier than learning transition functions -- would it be possible to find a reference for this? My personal opinion is that it should indeed be often the case, but that generally it is not true (e.g. imagine an MDP with a small transition matrix but an extremely stochastic reward function).

**Summary Of The Paper:**

The authors propose a framework for learning to synthesize a proxy models for usually non-learned components of an RL loop: namely transition dynamics and reward as defined in the interacting environment. They learn the parameters of these model by "meta-learning" onto the real environment with a learning agent (with slight detail variants depending on what the learning objective is), showing that not only can this setup lead to successfully learn the proxy parameters, but also improve learning performance for the agents.


**Summary Of The Review:**


This is a good paper. It contains excellent writing, good research, and makes for a great example of what an ICLR paper should look like. The research problem and proposed methods are interesting and well placed in the literature, and the experimental section is exhaustive.

---

> ### Author Response · Authors · 2021-11-13
> **Response to Reviewer pG56 (Part 1)**
>
> > First and foremost, let me start by saying that I really like this manuscript.
>
> We are very grateful for your feedback and your encouraging words. Thank you! We will not repeat the parts you liked but focus on parts you mention could be improved.
>
> \
> &nbsp;
>
> > The manuscript at times feels very dense, especially when looking at methodology and experimental details, and its understanding is greatly helped from the presence of a sizable appendix. It very much feels like the authors could have a feasibly split the paper into two, one on ES and a follow up on RN, providing more space to experimentally analyse ES and RN separately and make for a cleaner and easier review process.
>
> We are very grateful for your helpful feedback. We would like to take your suggestion to heart and try to reduce the overall density in the mentioned sections. In order to do this effectively, we would welcome concrete suggestions; in our perspective, splitting the paper into two, one on SEs and one on RNs, is not a preferable option since our conclusions and claims are often derived and justified from observations made on both SE and RN. Also, while we are very grateful that you feel there is enough material in the paper for two papers, other reviewers are rather asking for more results not less.
>
> \
> &nbsp;
>
> > Some of the hypotheses presented to justify the improved agent training feel relatively handwave-y. Consider for instance the following quote (emphasis mine):
> “Optimizing SEs for the maximum cumulative reward automatically yielded fast training of agents as a side-product, likely due to efficient synthetic state dynamics.”
>
> > It is implied that this meta-learning approach leads to generally discover better data-producing dynamics for the agents used in the system, but I wonder whether this is primarily due to the structure of the tasks employed in the experimental settings (as state space and transition dynamics of these simple control problems are generally not very informative, and the useful -- policy learning wise -- parts of these MDP are instead both easily discoverable via search and extremely informative).
>
> > In short, I wonder if we'd see similar improvements for tasks / environments where task difficulty is extremely affected by the emerging complexity of the environment, or whether we can actually show that this is a property of SEs / RNs when trained in this manner (which would be quite outstanding!). Although that said, the effecitveness of RNs do seem to indeed point to the latter hypothesis.
>
> Thank you for our feedback. We are uncertain whether we understand your point fully, especially with regard to the property of being “informative”. We believe the state space and transition dynamics of CartPole and Acrobot are informative in both a meta-learning and non-meta-learning setting as they can be purely described deterministically and analytically (e.g., do not rely on potentially approximate collision detection or contact simulation) and are bounded by clearly defined and deterministic termination and reward criteria (i.e. no stochasticity involved).
>
> If the assumption is that the beneficial data-producing dynamics arise from the simplicity of the used control problems and not from our meta-learning formulation, we argue that we would see similar benefits in a (non-meta-learning) supervised-learning or model-based RL setup. Fortunately, we have conducted an extensive experiment to study exactly this question. For the results, we kindly refer to Appendix D. Overall, we conclude from this study that the efficiency is very likely to arise from the meta-learning scheme and is challenging to reproduce through purely supervised-learning despite the simplicity of the task. As you pointed out, we think the RN results back up this claim as well.
>
> With this in mind, we nevertheless see high-complexity environments as an important step to justify our claim further. Despite our limited compute resources, we are preparing the launch of experiments on the complex HalfCheetah environment and given that these compute resources are sufficient, we are hopeful that we can present results at the end of the rebuttal period.
>
> Please let us know if we understood / addressed your point correctly and also if we should rephrase or pull relevant information from the appendix into the main manuscript to make this point clearer.
>
> \
> &nbsp;
>
> > Nits / Additional comments
> > Figure 2: the labels are inverted (dotted lines should be the thresholds).
>
> Thank you very much for pointing this out. We have fixed the labels and updated the manuscript.

---

> ### Author Response · Authors · 2021-11-13
> **Response to Reviewer pG56 (Part 2)**
>
> > Section 6 (but generally in many parts of the paper): it is claimed that learning rewards is generally easier than learning transition functions -- would it be possible to find a reference for this? My personal opinion is that it should indeed be often the case, but that generally it is not true (e.g. imagine an MDP with a small transition matrix but an extremely stochastic reward function).
>
> Unfortunately, we could not find such a reference but of course would be very happy to include it in the manuscript if any of the reviewers happen to know of one.
>
> We completely agree with you that one can construct counter-examples where reward learning is much more difficult than state learning. However, in practice, for instance, in scenarios where complex physical laws are simulated, it is generally much more difficult to come up with such counter-examples because the simulated dynamics are non-trivial. Moreover, when considering control or robotics applications, dynamics are often given and cannot be modified but rewards usually can be adapted to solve a problem.

---

### Official Review · Reviewer_1Zys · 2021-11-03

**Correctness:** 3
**Technical Novelty And Significance:** 3
**Empirical Novelty And Significance:** 3
**Recommendation:** 6
**Confidence:** 4

**Main Review:**

This is an interesting line of work and initial investigation. The exposition of the method and description of experiments are quite clear and complete. However, there are a few weaknesses.

The motivation for this work deserves more attention. The authors suggest that SE’s could be used for AutoML, iterating on algorithms in expensive environments like robotics, agent or task analysis, or agent pre-training. However, I’m not sure these are particularly well aligned with the authors’ statement of the problem, implementation, or experiments.
Most applications are predicated on the relative performance of various algorithms, when trained in the SE, being correlated with performance in the true environment.
However, the problem formulation in eqn (1) describes maximising performance absolutely.
This is approximated in Alg. 1 with a generic TrainAgent function that approximates the argmax_theta. But it seems the specific TrainAgent functions chosen are highly relevant to potential use cases – it is not a generic stand-in for an argmax.
The authors certainly understand the relevance of this, as they study generalisation of SE’s to different algorithms and their hyperparameters, and even address generalisation algorithmically by varying hyperparameters during training of the SEs. However, this is introduced as a brief comment in the experiment descriptions, rather than being central to the method.
It feels that to use SE’s for the potential applications discussed, it would make more sense to reformulate the problem statement to explicitly account for different inner-loop algorithms.
Further, simply maximising the final performance of all inner-loop algorithms seems quite limited when applications involve comparing different algorithms.
It might be interesting to algorithmically address this use-case by changing the fitness function to account for the relative performance of different algorithms.
However, it would also be useful to address this empirically given the current formulation of the objective, and the authors likely already have the data to do so, at least partially: does the performance of different HPs/algorithms *in the SEs* correlate with their performance on the true environment?

It may also be possible to motivate the work from a more traditional model-based RL point of view, which might be more closely aligned to the problem statement given in the paper. A closely related work is that of Nikishin et al [1] (who use implicit differentiation to address the bi-level optimisation). NB I don’t fault the authors for not citing this, I believe it is only available as preprint. They argue that models learned only to further optimisation of a policy (as in this work's problem statement) may perform better than traditional models when capacity is limited. However, the ES-based bi-level optimisation is expensive, so it may be difficult to make a strong case for this approach.

Another weakness of the study is the simplicity of the environments used. I understand the authors may have limited computational resources, but it is difficult to know how highly to weight an empirical feasibility study carried out on such small-scale tasks. In the absence of more compelling empirical findings, it is more critical to explore the algorithmic choices and motivations.

In this vein, I did appreciate the investigation of which inductive biases might help during reward learning. Overall, I find the case for RNs is perhaps more intuitive: dense-ifying rewards once in an expensive optimisation to allow fast iteration later. Using an additive potential form even guarantees that this will not change the optimal behaviour – sadly, it seems this type of RN did not outperform the baseline much in the experiments.

In the RN case optimising for the speed of training makes sense, but I still find the chosen objective strange. Why not optimise the area-under-curve returns for the full inner-loop? Clipping at a somewhat arbitrary solution threshold seems like it would result in suboptimal RNs (this appears to have occurred in the half-cheetah experiment in the appendix).

Minor comments:
 - In the SE experiments, it would be good to show the performance of the ‘fixed’ hyperparams evaluated with the same fixed hyperparams: this would show the ‘generalisation gap’ compared to evaluating with varied hyperparms as is currently shown.
 - Overall, many figures are a bit hard to parse. Try using more smoothing or subsampling for curves. For the barplots, maybe re-arrange them so the comparable bars (i.e. measuring the same metric, ‘steps’ or ‘episodes’, are next to each other).

[1] “Control-Oriented Model-Based Reinforcement Learning with Implicit Differentiation” Nikishin et al 2021

----------------------

Update: the authors have addressed many of my concerns, ran some additional useful experiments, and clarified their reasoning on several points in the overall discussion with reviewers. I still think there is a bit of a gap between the big-picture motivation for the work and the instantiation we see studied empirically; the authors are working on an empirical investigation to help make that connection. However, even in its absence, I am sufficiently positive about the paper overall now to lean towards acceptance -- I found the study interesting and creative.



**Summary Of The Paper:**

This paper proposes to learn neural environment models they term Synthetic Environments, such that planning in the model with model-free RL results in good performance in the real environment.
The authors study learning full models as well as reward-only models they call Reward Networks, investigating the feasibility of learning these models as well as their robustness to different inner-loop algorithms and hyperparameters.

**Summary Of The Review:**

While the work is interesting, I find the motivation a little strained at not well aligned with the problem statement or algorithmic instantiation. The empirical study is fairly clear but limited by the simplicity of the domains and the restriction largely to assessing feasibility rather than giving clearer indications of how the method could be used in practice.

---

> ### Author Response · Authors · 2021-11-13
> **Response to Reviewer 1Zys (Part 1)**
>
> > The motivation for this work deserves more attention. [...] Most applications are predicated on the relative performance of various algorithms, when trained in the SE, being correlated with performance in the true environment. However, the problem formulation in eqn (1) describes maximising performance absolutely. [...] Further, simply maximising the final performance of all inner-loop algorithms seems quite limited when applications involve comparing different algorithms
>
> Thank you for sharing this concern. We believe that there may be a misunderstanding. To clarify, we are not exchanging RL algorithms between outer loop iterations and thus we believe the TrainAgent function is a generic stand-in for an argmax. It is only *after* we have successfully optimized for an SE by executing Alg. 1 that we compare different RL algorithms on it. As you correctly pointed out in your Paper Summary, our work revolves around finding the SE models and in post-hoc evaluations we analyse transfer to different RL algorithms and their sensitivity to agent hyperparameter changes. Since these changing factors are only considered in the evaluation phase, we believe changing the problem formulation to account for these is not needed. However, changing the RL algorithms between outer loop iterations (also in addition to using pre-trained agents from previous outer loop iterations) is a very interesting idea that we already have on our future-work-stack.
>
> Did this address your concern and did we understand you correctly? We hope we were able to give some clarification on our motivation and experiment design and are happy to discuss further suggestions.
>
> \
> &nbsp;
>
> > The authors certainly understand the relevance of this, as they study generalisation of SE’s to different algorithms and their hyperparameters, and even address generalisation algorithmically by varying hyperparameters during training of the SEs. However, this is introduced as a brief comment in the experiment descriptions, rather than being central to the method.
>
> Thank you for pointing this out. We have integrated this information into the method section (Section 3; “Agents, Hyperparameters, and Sampling”) and updated the manuscript accordingly (changes highlighted in red color).
>
> \
> &nbsp;
>
> > Further, simply maximising the final performance of all inner-loop algorithms seems quite limited when applications involve comparing different algorithms. It might be interesting to algorithmically address this use-case by changing the fitness function to account for the relative performance of different algorithms.
>
> Thank you for sharing this idea with us. We believe this proposed new objective is interesting to study and may improve transferability to unseen agents. We will take this into consideration for our future work on learning SEs.
>
> \
> &nbsp;
> > However, it would also be useful to address this empirically given the current formulation of the objective, and the authors likely already have the data to do so, at least partially: does the performance of different HPs/algorithms in the SEs correlate with their performance on the true environment?
>
> We thank you for this interesting idea. Unfortunately, we do not have this data yet. We are currently preparing multiple experiments and will post an update here with the results once we have them.
>
>
> \
> &nbsp;
> > It may also be possible to motivate the work from a more traditional model-based RL point of view, which might be more closely aligned to the problem statement given in the paper. A closely related work is that of Nikishin et al [1] (who use implicit differentiation to address the bi-level optimisation). NB I don’t fault the authors for not citing this, I believe it is only available as preprint. They argue that models learned only to further optimisation of a policy (as in this work's problem statement) may perform better than traditional models when capacity is limited. However, the ES-based bi-level optimisation is expensive, so it may be difficult to make a strong case for this approach.
>
> Thank you for pointing out this related work and the suggestion to motivate our work through the lens of model-based RL (MBRL). Here, we would kindly point out that we provide an extensive discussion on the differences between our work and model-based RL in Appendix D. In summary, we decided against motivating our work based on MBRL because we see too many dissimilarities between both paradigms. As to the work of Nikishin et al. [1], we believe adopting ideas such as implicit differentiation from this paper is very valuable and provides interesting ideas for future work on gradient-based optimization for SEs.
>
> Regarding the SE model capacity, we have observed during hyperparameter optimization of the SE architecture, that its capacity did not seem to play a significant role. SE models seem to further the optimization of a policy irrelevant of whether the capacity was small or large.

---

> > ### Comment · Reviewer_1Zys · 2021-11-23
> > **Appreciate discussion, clarification of concern.**
> >
> > Clarification: I understand that you are not changing RL algorithms between outer-loop iterations. However, I think that not doing so undermines the motivations that you give for the work, some of which center on comparing different algorithms' performance in the SE, post-hoc. It's unclear why post-hoc performance of various algorithms would correspond to real-env performance, since only a single algorithm is used while learning the SE/RN. My challenge here is not to either the motivation per se, or the implementation, but rather the connection between them.
> >
> > I think the empirical study of performance correlations could go some way to bridging this gap.
> >
> > Thank you for your discussion of the other points and pointing me to the relevant part of the Appendix.

---

> ### Author Response · Authors · 2021-11-13
> **Response to Reviewer 1Zys (Part 2)**
>
> > [Simplicity of the environments used].
>
> Thank you for sharing this concern. We would like to emphasize that we not only propose a new method for learning environments but that we also propose the concept of creating SEs in the first place, and because of that, starting with simple environments is beneficial for several reasons, e.g.:
> * they allow for interpretability,
> * they are well-understood,
> * they are simple enough to be reproducible in the community without requiring large computational resources, and
> * they significantly reduce evaluation time.
>
> Having said that, we do agree that scalability to complex environments is an important issue that needs to be further investigated. In terms of SEs, we would like to highlight that we already chose environments with continuous state spaces where most of the related work uses simpler, discrete state spaces. For RNs, we would like to point out that we do already present results for the complex HalfCheetah environment.
> What is missing in our view is a complex environment, such as HalfCheetah, for SEs. Despite our limited compute resources, we are preparing the launch of experiments for training SEs on HalfCheetah, and given that the compute resources we get are sufficient, we are hopeful that we can present results at the end of the rebuttal period.
>
> Finally, we would like you to consider that most of the related work focussed on similarly or even less complex environments, such as grid worlds. For this reason, we believe that our empirical findings are very compelling. Not only do we show the feasibility of our method, but also report valuable insights into the learned SEs in terms of efficiency, agent transfer and an analysis of the distribution of synthetic states and rewards.
>
> We hope we were able to address your concern and we would be very happy to discuss more about this point.
>
> \
> &nbsp;
>
> > In the RN case optimising for the speed of training makes sense, but I still find the chosen objective strange. Why not optimise the area-under-curve returns for the full inner-loop? Clipping at a somewhat arbitrary solution threshold seems like it would result in suboptimal RNs (this appears to have occurred in the half-cheetah experiment in the appendix).
>
> Thank you for opening the discussion about the RN objective. Regarding the arbitrary solution thresholds, we would like to point out that the solved-reward thresholds were not arbitrarily chosen but taken from the widely adopted Gym competition leaderboard setups (reported here: https://github.com/openai/gym/wiki/Table-of-environments)
>
> In terms of the objective, we agree with you that this detail may be confusing to readers. Frankly speaking, the choice of this objective is not greatly motivated. We implemented this objective after we noticed the efficiency improvements from SEs. To test whether RNs could also yield such efficiency improvements, we chose to directly optimize for efficiency. Another reason for optimizing for efficiency directly was that the real environment is involved in the RN case and we pursued the goal to reduce real environment interactions.
>
> Nevertheless, we now conducted a small study to contrast different objectives in the RN case and made the results available here (results reported across 6 pages):
>
> https://docs.google.com/document/d/1LLshom6G8gr-Si6-RKPYfnhHM7BYrCeKhkwEJn2BTrk/edit?usp=sharing
>
>
> In these results, next to the area-under-the-curve (AUC) objective you mentioned, we report results for the RN objective that was used in the paper (denoted as “Reward Threshold”), as well as for the reward maximization objective (denoted as “max reward”). The overall conclusion from these results is that an AUC objective generally yields similar results as for the reward threshold objective, but in some cases it seems to reduce efficiency a bit. However, the overall significant efficiency gains remain. Moreover, in contrast to SEs, efficiency improvements in the RN case seem to arise more often when explicitly optimized for it (see Figures under “AUC” or “reward threshold”) while maximizing reward (only tested for CartPole and Cliff environments) leads to efficiency improvements only in the CartPole but not in the Cliff environment.
>
> Did we address this concern appropriately? Again, we would be happy to discuss further; we also plan to include the results of the above objective study in the manuscript.

---

> > ### Comment · Reviewer_1Zys · 2021-11-23
> > **Thanks**
> >
> > Regarding simplicity of environments; I do think there is information to take away from experiments on these environments. I disagree that discrete gridworlds are necessarily any simpler than simple continuous environments. But, I regard this factor as a minor issue overall; and believe there is a good quality paper to be written within the computational budget you have available.
> >
> > Regarding the RN objective: I'm aware the solution thresholds are established in gym, but still find them quite arbitrary regardless (of course, someone else's arbitrary decision is better than yours for unbiased research, but still can be unsatisfactory). So thank you for investigating this suggestion more thoroughly.
> >
> > Correct me if I'm interpreting the results incorrectly, but it seems that in particular the non-potential RNs are working better on HalfCheetah with the AUC objective? This would make sense to me since the non-potential RNs would be the ones most affected by a thresholded objective, and the more complex Cheetah may be the most sensitive to this factor.
> >
> > Regardless, I believe this is a useful addition to the paper and will make the discussion of RNs feel a bit better connected to the overall motivation (by not tying them so closely to an objective which is so different from the SE one). This may also be of interest to reviewer Q4vZ.

---

> ### Author Response · Authors · 2021-11-13
> **Response to Reviewer 1Zys (Part 3)**
>
> > Minor comments:
> > * In the SE experiments, it would be good to show the performance of the ‘fixed’ hyperparams evaluated with the same fixed hyperparams: this would show the ‘generalisation gap’ compared to evaluating with varied hyperparms as is currently shown.
> > * Overall, many figures are a bit hard to parse. Try using more smoothing or subsampling for curves. For the barplots, maybe re-arrange them so the comparable bars (i.e. measuring the same metric, ‘steps’ or ‘episodes’, are next to each other).
>
>
> Thank you for pointing these out. Regarding the generalization gap, we will launch an experiment and report back once we have the results for it. With reference to the suggestion of improving the curve plots, we will also look into that. Regarding the barplots, we will test out your suggestion for re-arranging the bars and report back to you.

---

> ### Author Response · Authors · 2021-11-16
> **Experiment Update for Reviewer 1Zys (Generalization Gap)**
>
> Dear reviewer,
>
> We would like to report the CartPole results of the experiment you suggested with regards to the generalization gap and are happy to discuss them here. The results can be accessed in the following Google Doc: https://docs.google.com/document/d/1SItWlHtV4qGi6JfuuHnGJpi0aY_oI2mlld7qmDBuOAM/edit?usp=sharing
>
> We used the fixed agent hyperparameters (reported in Appendix, Table 3) that were found for the SE optimization and repeated the evaluation procedure reported in the paper (Section 5.1, “Evaluating the Performance of SE-trained Agents”) with those fixed hyperparameters.
>
> One can see that these agent hyperparameters lead to high scores under the real environment baseline and lower performance on the SEs when comparing them against the results in the paper. We believe the reason for high scores in the real environment stems from the fact that we executed our BOHB hyperparameter optimization with the goal that agents, in the end, would perform well in the real environment (after being trained on an SE). However, these hyperparameters seem to also result in needing more episodes than the default (sampled) hyperparameters. This may stem from the lower learning rates in the fixed HP case compared to sampled learning rates from Table 1.
>
> Also, again we can observe that performance is higher when training agents on the SEs optimized with varying hyperparameters. We think this hyperparameter variation during training regularizes the SEs for higher generalization which might be beneficial on CartPole (for example, by coping better with the random initial start position of the pole). Overall, our results indicate a noticeable generalization gap of an average reward of ~50 when comparing agents trained on the SEs with the fixed and the varying hyperparameters.
>
> Do you feel we have addressed your experiment suggestion sufficiently enough?

---

> > ### Comment · Reviewer_1Zys · 2021-11-23
> > **Thanks**
> >
> > Appreciate this additional experiment. I guess it would probably be even more informative if the fixed hyperparams were better-performing in this setting, but I understand the motivation for choosing the fixed HPs that came from your optimization procedure.
> >
> > Although the signal is a bit weak as a result, it at least indicates a bit more clearly what's going on in this case!

---

> ### Author Response · Authors · 2021-11-20
> **Performance Correlation Experiment**
>
> > Reviewer 1Zys: However, it would also be useful to address this empirically given the current formulation of the objective, and the authors likely already have the data to do so, at least partially: does the performance of different HPs/algorithms in the SEs correlate with their performance on the true environment?
>
> > Paper Authors: We thank you for this interesting idea. Unfortunately, we do not have this data yet. We are currently preparing multiple experiments and will post an update here with the results once we have them.
>
> Dear reviewer,
>
> We would like to quickly give a status update on the correlation experiment that you suggested. We have launched the experiment runs for DDQN and DuelingDDQN and are hopeful that we can evaluate and post the results by tomorrow (Sunday).
>
> Also, did you already find time to look at the generalization gap experiment results that we posted on Nov 16 (see below)? We would be very happy to receive feedback and would be looking forward to a fruitful discussion about it.
>
> Thank you.

---

> > ### Comment · Reviewer_1Zys · 2021-11-23
> > **Appreciate followup, considering overall review**
> >
> > I'd like to thank the authors for their engagement with the reviews. The additional results and clarifications definitely alleviate some of my concerns. I am also considering the points raised in the other reviews, and will likely amend my score shortly.
> >
> > I'd also like to echo another reviewer's comment about density and volume of information, and encourage the authors to think about streamlining the presentation further; perhaps aggregating results together or de-emphasising less-informative experiments to help the empirical section have clearer take-aways where possible.

---

> > > ### Author Response · Authors · 2021-11-27
> > > **Following up on the follow-up :-)**
> > >
> > > We thank you in return for your response and the very interesting and lively discussion. We believe this is what a rebuttal should look like and highly appreciate the effort to provide us with this very detailed feedback. Hopeful that we have addressed and mitigated your initial concerns, frankly, we would like to ask you if you could increase your score (or, alternatively, we would be glad for any points we should address that would raise your score).
> > >
> > > In the following, we address your remaining questions:
> > > \
> > > &nbsp;
> > >
> > > > Appreciate this additional experiment. I guess it would probably be even more informative if the fixed hyperparams were better-performing in this setting, but I understand the motivation for choosing the fixed HPs that came from your optimization procedure.
> > > Although the signal is a bit weak as a result, it at least indicates a bit more clearly what's going on in this case!
> > >
> > > We also think that these results are interesting and will include these together with the discussion in the updated version of the paper.
> > >
> > > \
> > > &nbsp;
> > >
> > >
> > > > Regarding simplicity of environments; I do think there is information to take away from experiments on these environments. I disagree that discrete gridworlds are necessarily any simpler than simple continuous environments. But, I regard this factor as a minor issue overall; and believe there is a good quality paper to be written within the computational budget you have available.
> > >
> > > > Correct me if I'm interpreting the results incorrectly, but it seems that in particular the non-potential RNs are working better on HalfCheetah with the AUC objective? This would make sense to me since the non-potential RNs would be the ones most affected by a thresholded objective, and the more complex Cheetah may be the most sensitive to this factor.
> > >
> > > Yes, we completely agree with your observation and also arrive at the same conclusion. Thank you for pointing this out. We will add this to the discussion of the HalfCheetah experiment.
> > >
> > > \
> > > &nbsp;
> > >
> > > > Regardless, I believe this is a useful addition to the paper and will make the discussion of RNs feel a bit better connected to the overall motivation (by not tying them so closely to an objective which is so different from the SE one). This may also be of interest to reviewer Q4vZ.
> > >
> > > Indeed, we also think this is a very useful addition to the paper which makes the paper more sound by better connecting RNs to the motivation. Consequently, we would like to replace the existing RN figures in the main paper by the AUC figures that we presented in the Google Doc. and include a discussion on different RN objectives to the appendix.
> > >
> > > \
> > > &nbsp;
> > >
> > > > Clarification: I understand that you are not changing RL algorithms between outer-loop iterations. However, I think that not doing so undermines the motivations that you give for the work, some of which center on comparing different algorithms' performance in the SE, post-hoc. It's unclear why post-hoc performance of various algorithms would correspond to real-env performance, since only a single algorithm is used while learning the SE/RN. My challenge here is not to either the motivation per se, or the implementation, but rather the connection between them.
> > >
> > > > I think the empirical study of performance correlations could go some way to bridging this gap.
> > >
> > > > Thank you for your discussion of the other points and pointing me to the relevant part of the Appendix.
> > >
> > >
> > > Thank you for clarifying our misunderstanding. We agree that the performance correlation experiment would bridge this gap. Unfortunately, this experiment is still running. Measuring the correlations with the stochasticity in RL agents and the real environments requires a large number of evaluations to allow for solid conclusions with significant correlations. Given paper acceptance, we are confident that this experiment result will be available before the camera-ready deadline.
> > >
> > > Nonetheless, we will include this in our future synthetic environment research.

---

### Author Response · Authors · 2021-11-13
**Common Comment to Reviewers**

We are very grateful for the helpful feedback and would like to thank the reviewers for their time and effort, and hope that they and their loved ones are safe and healthy. We will focus on each of the reviewer’s concerns individually and are looking forward to a lively discussion.

---

### Decision · Program_Chairs · 2022-01-20

**Decision:**

Accept (Poster)

**Comment:**

This paper proposes a new method for generating synthetic environments and reward networks for reinforcement learning tasks. This happens as a nested process: policies are learned in an inner loop, and environments are evolved in an outer loop. The environment representation is quite simple: the parameters of an MDP. Similarly, the reward networks are simply neural networks. Results show that the the learned environments and reward networks are reasonably good at decreasing policy training time by RL. The proposed method appears to be simple and quite general, and it would be interesting to see how it scales up to more complex environment representations.

The discussion around the paper centered on understanding various details of the method, and on the quality of the results. The reviewers generally agree that the paper is easy to read, and vary in their assessment of the significance of the results. It was pointed out that the generated environments are not necessarily similar to the base tasks, but it was nowhere claimed in the paper that they were. (In fact, it could be argued that the dissimilarity makes the method more interesting, given the good results of policy training.)

I'm happy to recommend the paper for poster acceptance. If the results would have been more impressive, it could have been accepted for a more prominent presentation form; however, I believe that the method can yield better results in the future with more sophisticated environment representations.